# Voyager: An Open-Ended Embodied Agent with Large Language Models

**Guanzhi Wang**[1][2]✉, **Yuqi Xie**[3], **Yunfan Jiang**[4]*, **Ajay Mandlekar**[1]*,
**Chaowei Xiao**[1][5], **Yuke Zhu**[1][3], **Linxi "Jim" Fan**[1]†✉, **Anima Anandkumar**[1][2]†
[1]**NVIDIA**, [2]**Caltech**, [3]**UT Austin**, [4]**Stanford**, [5]**UW Madison**
*__Equal contribution__   †__Equal advising__   ✉ **Corresponding authors**
https://voyager.minedojo.org

Reviewed on OpenReview: https://openreview.net/forum?id=ehfRiF0R3a

## Abstract

We introduce VOYAGER, the first LLM-powered embodied lifelong learning agent in Minecraft that continuously explores the world, acquires diverse skills, and makes novel discoveries without human intervention. VOYAGER consists of three key components: 1) an automatic curriculum that maximizes exploration, 2) an ever-growing skill library of executable code for storing and retrieving complex behaviors, and 3) a new iterative prompting mechanism that incorporates environment feedback, execution errors, and self-verification for program improvement. VOYAGER interacts with GPT-4 via blackbox queries, which bypasses the need for model parameter fine-tuning. The skills developed by VOYAGER are temporally extended, interpretable, and compositional, which compounds the agent's abilities rapidly and alleviates catastrophic forgetting. Empirically, VOYAGER shows strong in-context lifelong learning capability and exhibits exceptional proficiency in playing Minecraft. It obtains 3.3× more unique items, travels 2.3× longer distances, and unlocks key tech tree milestones up to 15.3× faster than prior SOTA. VOYAGER is able to utilize the learned skill library in a new Minecraft world to solve novel tasks from scratch, while other techniques struggle to generalize.

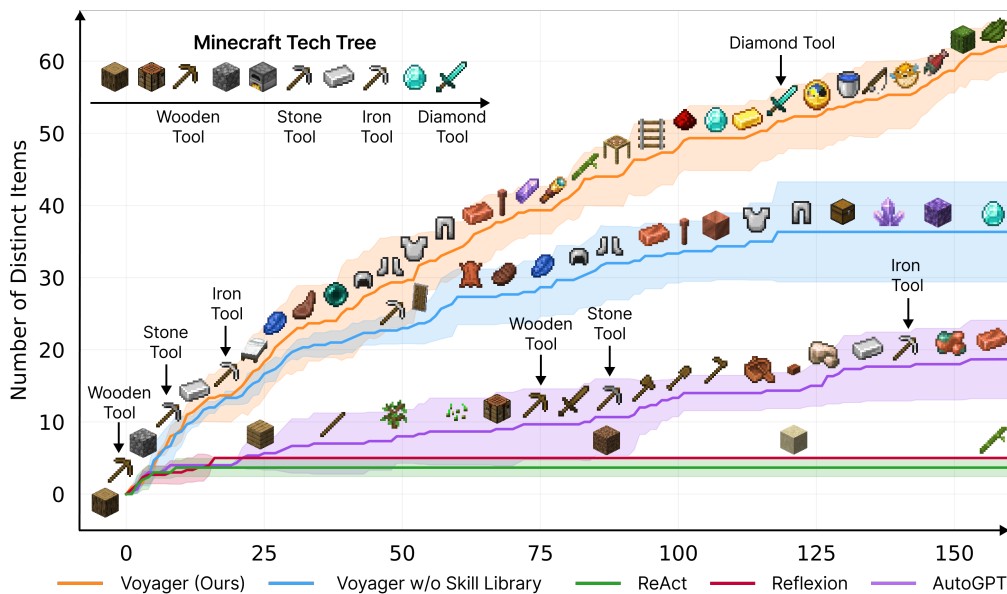

Figure 1: VOYAGER discovers new Minecraft items and skills continually by self-driven exploration, significantly outperforming the baselines. X-axis denotes the number of prompting iterations. The shaded region represents the standard deviation of the items acquired by each agent over three experimental runs. Voyager is the only method that can obtain a diamond tool in one of the three runs.

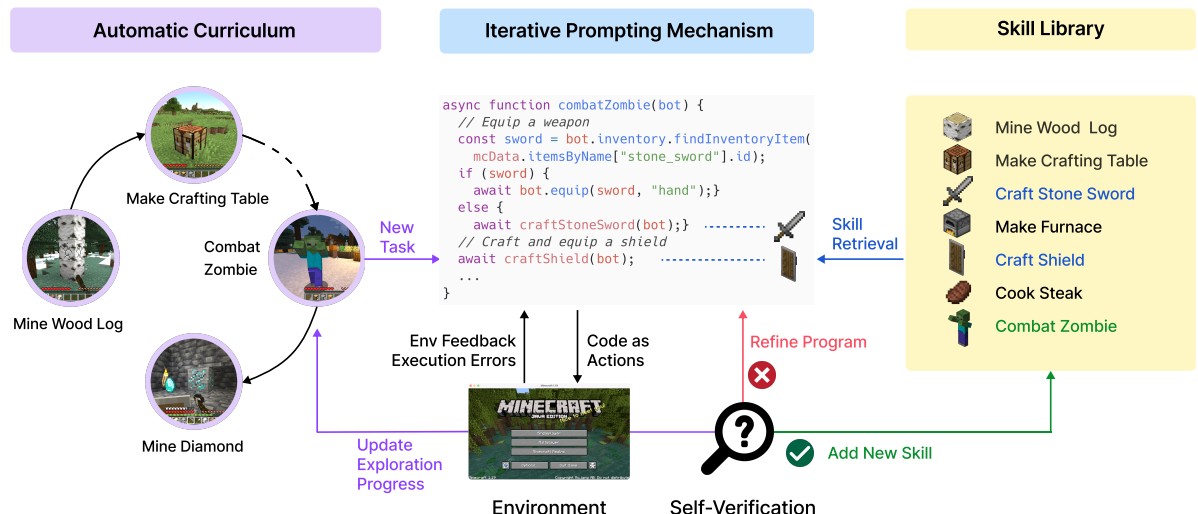

Figure 2: VOYAGER consists of three key components: an automatic curriculum for open-ended exploration, a skill library for increasingly complex behaviors, and an iterative prompting mechanism that uses code as action space.

# 1 Introduction

Building generally capable embodied agents that continuously explore, plan, and develop new skills in open-ended worlds is a grand challenge for the AI community (Kolve et al., 2017; Savva et al., 2019; Zhu et al., 2020; Xia et al., 2019; Shen et al., 2020). Classical approaches employ reinforcement learning (RL) (Kober et al., 2013; Arulkumaran et al., 2017) and imitation learning (Baker et al., 2022; Team et al., 2021; Vinyals et al., 2019) that operate on primitive actions, which could be challenging for systematic exploration (Ecoffet et al., 2019; Huizinga & Clune, 2022; Wang et al., 2020; Kanitscheider et al., 2021; Dennis et al., 2020), interpretability (Liang et al., 2022; Sun et al., 2020; Zhao et al., 2021), and generalization (Jiang et al., 2022; Shridhar et al., 2021; Fan et al., 2021). Recent advances in large language model (LLM) based agents harness the world knowledge encapsulated in pre-trained LLMs to generate consistent action plans or executable policies (Liang et al., 2022; Singh et al., 2022; Jiang et al., 2022). They are applied to embodied tasks like games and robotics (Fan et al., 2022; Zeng et al., 2022; Ahn et al., 2022; Huang et al., 2022b;a), as well as NLP tasks without embodiment (Richards, 2023; Yao et al., 2022; Shinn et al., 2023). However, these agents are not lifelong learners that can progressively acquire, update, accumulate, and transfer knowledge over extended time spans (Parisi et al., 2019; Wang et al., 2023a).

Let us consider Minecraft as an example. Unlike most other games studied in AI (Mnih et al., 2013; OpenAI et al., 2019; Vinyals et al., 2019), Minecraft does not impose a predefined end goal or a fixed storyline but rather provides a unique playground with endless possibilities (Fan et al., 2022). Minecraft requires players to explore vast, procedurally generated 3D terrains and unlock a tech tree using gathered resources. Human players typically start by learning the basics, such as mining wood and cooking food, before advancing to more complex tasks like combating monsters and crafting diamond tools. We argue that an effective lifelong learning agent should have similar capabilities as human players: (1) **propose suitable tasks** based on its current skill level and world state, e.g., learn to harvest sand and cactus before iron if it finds itself in a desert rather than a forest; (2) **refine skills** based on environmental feedback and **commit mastered skills to memory** for future reuse in similar situations (e.g. fighting zombies is similar to fighting spiders); (3) **continually explore the world** and seek out new tasks in a self-driven manner.

Towards these goals, we introduce VOYAGER, the first *LLM-powered embodied lifelong learning agent* to drive exploration, master a wide range of skills, and make new discoveries continually without human intervention in Minecraft. VOYAGER is made possible through three key modules (Fig. 2): 1) an **automatic curriculum** that maximizes exploration; 2) a **skill library** for storing and retrieving complex behaviors; and 3) a new

**iterative prompting mechanism** that generates executable code for embodied control. We opt to use code as the action space instead of low-level motor commands because programs can naturally represent temporally extended and compositional actions (Liang et al., 2022; Singh et al., 2022), which are essential for many long-horizon tasks in Minecraft. VOYAGER interacts with a blackbox LLM (GPT-4 (OpenAI, 2023)) through prompting and in-context learning (Wei et al., 2022a; Brown et al., 2020; Raffel et al., 2020). Our approach bypasses the need for model parameter access and explicit gradient-based training or finetuning.

More specifically, VOYAGER attempts to solve progressively harder tasks proposed by the **automatic curriculum**, which takes into account the exploration progress and the agent's state. The curriculum is generated by GPT-4 based on the overarching goal of "discovering as many diverse things as possible". This approach can be perceived as an in-context form of *novelty search* (Eysenbach et al., 2019; Conti et al., 2018). VOYAGER incrementally builds a **skill library** by storing the action programs that help solve a task successfully. Each program is indexed by the embedding of its description, which can be retrieved in similar situations in the future. Complex skills can be synthesized by *composing* simpler programs, which compounds VOYAGER's capabilities rapidly over time and alleviates catastrophic forgetting in other continual learning methods (Parisi et al., 2019; Wang et al., 2023a).

However, LLMs struggle to produce the correct action code consistently in one shot (Chen et al., 2021a). To address this challenge, we propose an **iterative prompting mechanism** that: (1) executes the generated program to obtain observations from the Minecraft simulation (such as inventory listing and nearby creatures) and error trace from the code interpreter (if any); (2) incorporates the feedback into GPT-4's prompt for another round of code refinement; and (3) repeats the process until a self-verification module confirms the task completion, at which point we commit the program to the skill library (e.g., `craftStoneShovel()` and `combatZombieWithSword()`) and query the automatic curriculum for the next milestone (Fig. 2).

Empirically, VOYAGER demonstrates strong **in-context lifelong learning** capabilities. It can construct an ever-growing skill library of action programs that are reusable, interpretable, and generalizable to novel tasks. We evaluate VOYAGER systematically against other LLM-based agent techniques (e.g., ReAct (Yao et al., 2022), Reflexion (Shinn et al., 2023), AutoGPT (Richards, 2023)) in MineDojo (Fan et al., 2022), an open-source Minecraft AI framework. VOYAGER outperforms prior SOTA by obtaining 3.3× more unique items, unlocking key tech tree milestones up to 15.3× faster, and traversing 2.3× longer distances. We further demonstrate that VOYAGER is able to utilize the learned skill library in a new Minecraft world to solve novel tasks from scratch, while other methods struggle to generalize.

## 2 Method

VOYAGER consists of three novel components: (1) an automatic curriculum (Sec. 2.1) that suggests objectives for open-ended exploration, (2) a skill library (Sec. 2.2) for developing increasingly complex behaviors, and (3) an iterative prompting mechanism (Sec. 2.3) that generates executable code for embodied control. The pseudocode of VOYAGER algorithm is shown in Pseudocode 1. Full prompts are presented in Appendix, Sec. A.

### 2.1 Automatic Curriculum

Embodied agents encounter a variety of objectives with different complexity levels in open-ended environments. An automatic curriculum offers numerous benefits for open-ended exploration, ensuring a challenging but manageable learning process, fostering curiosity-driven intrinsic motivation for agents to learn and explore, and encouraging the development of general and flexible problem-solving strategies (Wang et al., 2019; Portelas et al., 2020; Forestier et al., 2022). Our automatic curriculum capitalizes on the internet-scale knowledge contained within GPT-4 by prompting it to provide a steady stream of new tasks or challenges. The curriculum unfolds in a bottom-up fashion, allowing for considerable adaptability and responsiveness to the exploration progress and the agent's current state (Fig. 3). As VOYAGER progresses to harder self-driven goals, it naturally learns a variety of skills, such as "mining a diamond".

The input prompt to GPT-4 consists of several components:

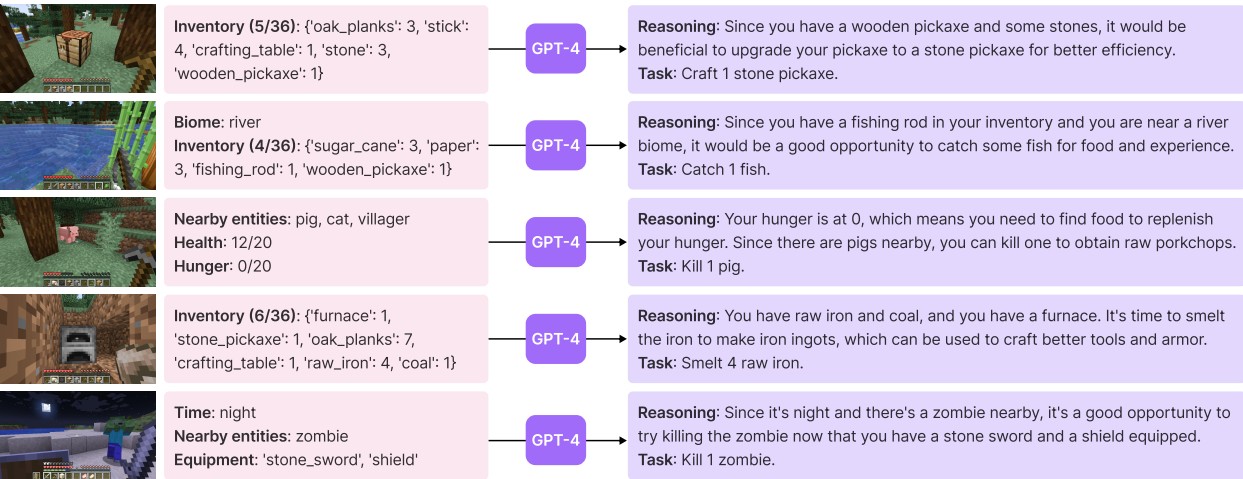

Figure 3: Tasks proposed by the automatic curriculum. We only display the partial prompt for brevity. See Appendix, Sec. A.2 for the full prompt structure.

(1) **Directives encouraging diverse behaviors and imposing constraints**, such as "`My ultimate goal is to discover as many diverse things as possible ... The next task should not be too hard since I may not have the necessary resources or have learned enough skills to complete it yet.`";

(2) **The agent's current state**, including inventory, equipment, nearby blocks and entities, biome, time, health and hunger bars, and position;

(3) **Previously completed and failed tasks**, reflecting the agent's current exploration progress and capabilities frontier;

(4) **Additional context**: We also leverage GPT-3.5 to self-ask questions based on the agent's current state and exploration progress and self-answer questions. We opt to use GPT-3.5 instead of GPT-4 for standard NLP tasks due to budgetary considerations.

## 2.2 Skill Library

With the automatic curriculum consistently proposing increasingly complex tasks, it is essential to have a skill library that serves as a basis for learning and evolution. Inspired by the generality, interpretability, and universality of programs (Ellis et al., 2020), we represent each skill with executable code that scaffolds temporally extended actions for completing a specific task proposed by the automatic curriculum.

The input prompt to GPT-4 consists of the following components:

(1) **Guidelines for code generation**, such as "`Your function will be reused for building more complex functions. Therefore, you should make it generic and reusable.`";

(2) **Control primitive APIs, and relevant skills** retrieved from the skill library, which are crucial for in-context learning (Wei et al., 2022a; Brown et al., 2020; Raffel et al., 2020) to work well;

(3) **The generated code from the last round, environment feedback, execution errors, and critique**, based on which GPT-4 can self-improve (Sec. 2.3);

(4) **The agent's current state**, including inventory, equipment, nearby blocks and entities, biome, time, health and hunger bars, and position;

(5) **Chain-of-thought prompting** (Wei et al., 2022b) to do reasoning before code generation.

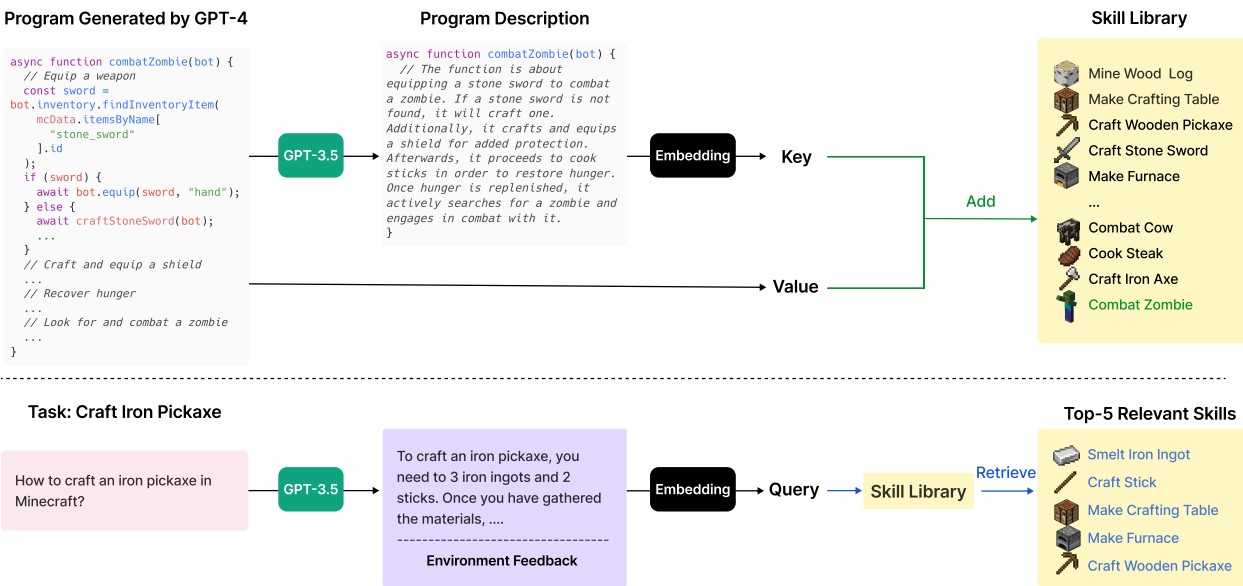

Figure 4: Skill library. **Top: Adding a new skill.** Each time GPT-4 generates and verifies a new skill, we add it to the skill library, represented by a vector database. The key is the embedding vector of the program description (generated by GPT-3.5), while the value is the program itself. **Bottom: Skill retrieval.** When faced with a new task proposed by the automatic curriculum, we first leverage GPT-3.5 to generate a general suggestion for solving the task, which is combined with environment feedback as the query context. Subsequently, we perform querying to identify the top-5 relevant skills.

We iteratively refine the program through a novel iterative prompting mechanism (Sec. 2.3), incorporate it into the skill library as a new skill, and index it by the embedding of its description (Fig. 4, top). For skill retrieval, we query the skill library with the embedding of self-generated task plans and environment feedback (Fig. 4, bottom). By continuously expanding and refining the skill library, VOYAGER can learn, adapt, and excel in a wide spectrum of tasks, consistently pushing the boundaries of its capabilities in the open world.

## 2.3 Iterative Prompting Mechanism

We introduce an iterative prompting mechanism for self-improvement through three types of feedback:

(1) **Environment feedback**, which illustrates the intermediate progress of program execution (Fig. 5, left). For example, "I cannot make an iron chestplate because I need: 7 more iron ingots" highlights the cause of failure in crafting an iron chestplate. We use `bot.chat()` inside control primitive APIs to generate environment feedback and prompt GPT-4 to use this function as well during code generation;

(2) **Execution errors** from the program interpreter that reveal any invalid operations or syntax errors in programs, which are valuable for bug fixing (Fig. 5, right);

(3) **Self-verification for checking task success.** Instead of manually coding success checkers for each new task proposed by the automatic curriculum, we instantiate another GPT-4 agent for self-verification. By providing VOYAGER's current state and the task to GPT-4, we ask it to act as a critic (Mnih et al., 2016; Schulman et al., 2017; Lillicrap et al., 2016) and inform us whether the program achieves the task. In addition, if the task fails, it provides a critique by suggesting how to complete the task (Fig. 6). Hence, our self-verification is more comprehensive than self-reflection (Shinn et al., 2023) by both checking success and reflecting on mistakes.

**Environment Feedback**

I cannot make stick because I need: 2 more planks
I cannot make stone_shovel because I need: 2 more stick

**Execution Error**

```
throw new Error(`No item named ${name}`);
No item named acacia_axe
at line 18:await craftItem(bot, "acacia_axe", 1);
```

GPT-4

GPT-4

```
async function craftStoneShovelWithTable(bot) {
    // If not enough cobblestone, mine cobblestone
    ...
+   // If not enough sticks, check if there are
+   // enough planks in the inventory
    if (sticksCount < 2) {
+       const planksCount = bot.inventory.count(
+           mcData.itemsByName.acacia_planks.id);
+       if (planksCount < 2) {
+           // Collect or craft planks
+           await mineBlock(bot, "acacia_log", 1);
+           await craftItem(bot, "acacia_planks", 1);
+       }
+       // Craft sticks using planks
        await craftItem(bot, "stick", 1);}
    await craftItem(bot, "stone_shovel", 1);
}
```

```
-async function craftAcaciaAxe(bot) {
+async function craftWoodenAxe(bot) {
    // Craft anacia planks and sticks
    ...

    // Place the crafting table near the bot
    ...

-   // Craft an acacia axe using 3 acacia planks
-   // and 2 sticks
-   await craftItem(bot, "acacia_axe", 1);
-   bot.chat("Acacia axe crafted.");
+   // Craft a wooden axe using 3 acacia planks
+   // and 2 sticks
+   await craftItem(bot, "wooden_axe", 1);
+   bot.chat("Wooden axe crafted.");
}
```

Figure 5: **Left: Environment feedback.** GPT-4 realizes it needs 2 more planks before crafting sticks. **Right: Execution error.** GPT-4 realizes it should craft a wooden axe instead of an acacia axe since there is no acacia axe in Minecraft. We only display the partial prompt for brevity. The full prompt structure for code generation is in Appendix, Sec. A.3.

During each round of code generation, we execute the generated program to obtain environment feedback and execution errors from the code interpreter, which are incorporated into GPT-4's prompt for the next round of code refinement. This iterative process repeats until self-verification validates the task's completion, at which point we add this new skill to the skill library and ask the automatic curriculum for a new objective (Fig. 2). If the agent gets stuck after 4 rounds of code generation, then we query the curriculum for another task. This iterative prompting approach significantly improves program synthesis for embodied control, enabling VOYAGER to continuously acquire diverse skills without human intervention.

## 3 Experiments

### 3.1 Experimental Setup

We leverage OpenAI's `gpt-4-0314` (OpenAI, 2023) and `gpt-3.5-turbo-0301` (chatgpt) APIs for text completion, along with `text-embedding-ada-002` (embedding) API for text embedding. We set all temperatures to 0 except for the automatic curriculum, which uses temperature = 0.1 to encourage task diversity. Our simulation environment is built on top of MineDojo (Fan et al., 2022) and leverages Mineflayer (PrismarineJS, 2013) JavaScript APIs for motor controls. See Appendix, Sec. B.1 for more details.

### 3.2 Baselines

Because there is no LLM-based agents that work out of the box for Minecraft, we make our best effort to select a number of representative algorithms as baselines. These methods are originally designed only for NLP tasks without embodiment, therefore we have to re-interpret them to be executable in MineDojo and compatible with our experimental setting:

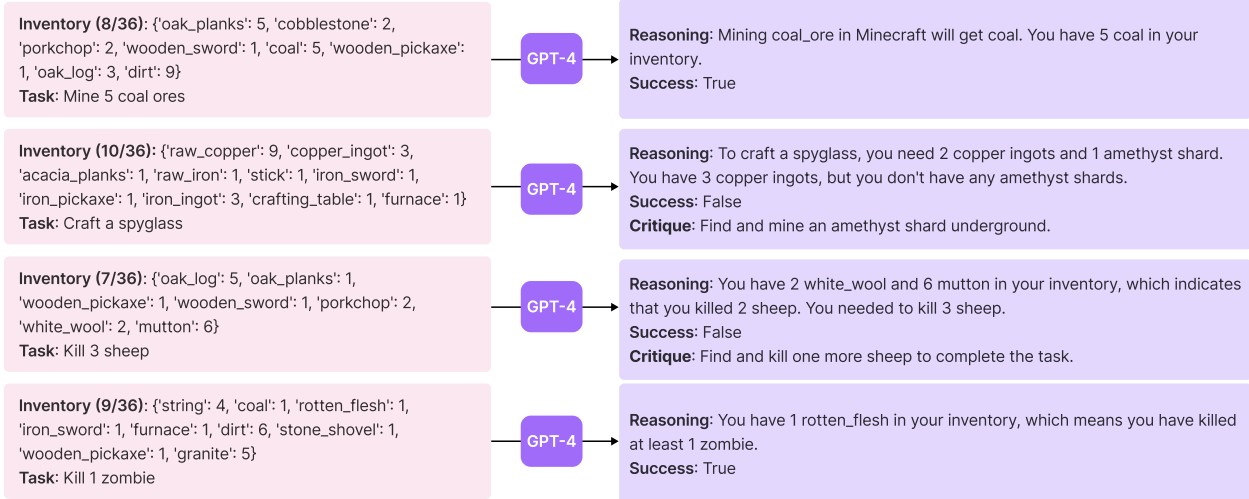

Figure 6: Self-verification examples. We only display the partial prompt for brevity. See Appendix, Sec. A.4 for the full prompt structure.

Pseudocode 1: VOYAGER algorithm.

```
def voyager(
    environment,  # environment that uses code as action space
    curriculum_agent,  # curriculum agent for proposing the next task
    action_agent,  # action agent for code generation
    critic_agent,  # critic agent for self-verification
    skill_manager,  # skill manager for adding new skills and skill retrieval
):
    agent_state = environment.reset()
    while True:
        exploration_progress = (
            curriculum_agent.get_exploration_progress(
                curriculum_agent.get_completed_tasks(),
                curriculum_agent.get_failed_tasks(),
            )
        )
        task = curriculum_agent.propose_next_task(
            agent_state, exploration_progress
        )
        code = None
        environment_feedback = None
        execution_errors = None
        critique = None
        success = False
        # try at most 4 rounds before moving on to the next task
        for i in range(4):
            skills = skill_manager.retrieve_skills(
                task, environment_feedback
            )
            code = action_agent.generate_code(
                task,
                code,
                environment_feedback,
                execution_errors,
                critique,
                skills,
            )
            (
                agent_state,
                environment_feedback,
                execution_errors,
            ) = environment.step(code)
```

```
        success , critique = critic_agent.check_task_success (
            task , agent_state
        )
        if success :
            break
    if success :
        skill_manager.add_skill ( code )
        curriculum_agent.add_completed_task ( task )
    else :
        curriculum_agent.add_failed_task ( task )
```

**ReAct** (Yao et al., 2022) uses chain-of-thought prompting (Wei et al., 2022b) by generating both reasoning traces and action plans with LLMs. We provide it with our environment feedback and the agent states as observations.

**Reflexion** (Shinn et al., 2023) is built on top of ReAct (Yao et al., 2022) with self-reflection to infer more intuitive future actions. We provide it with execution errors and our self-verification module.

**AutoGPT** (Richards, 2023) is a popular software tool that automates NLP tasks by decomposing a high-level goal into multiple subgoals and executing them in a ReAct-style loop. We re-implement AutoGPT by using GPT-4 to do task decomposition and provide it with the agent states, environment feedback, and execution errors as observations for subgoal execution. Compared with VOYAGER, AutoGPT lacks the skill library for accumulating knowledge, self-verification for assessing task success, and automatic curriculum for open-ended exploration.

Note that we do not directly compare with prior methods that take Minecraft screen pixels as input and output low-level controls (Nottingham et al., 2023; Cai et al., 2023; Wang et al., 2023b). It would not be an apple-to-apple comparison, because we rely on the high-level Mineflayer (PrismarineJS, 2013) API to control the agent. Our work's focus is on pushing the limits of GPT-4 for lifelong embodied agent learning, rather than solving the 3D perception or sensorimotor control problems. VOYAGER is orthogonal and can be combined with gradient-based approaches like VPT (Baker et al., 2022) as long as the controller provides a code API. We make a system-level comparison between VOYAGER and prior Minecraft agents in Table. A.2.

### 3.3 Evaluation Results

We systematically evaluate VOYAGER and baselines on their exploration performance, tech tree mastery, map coverage, and zero-shot generalization capability to novel tasks in a new world.

**Significantly better exploration.** Results of exploration performance are shown in Fig. 1. VOYAGER's superiority is evident in its ability to consistently make new strides, discovering 63 unique items within 160 prompting iterations, 3.3× many novel items compared to its counterparts. On the other hand, AutoGPT lags considerably in discovering new items, while ReAct and Reflexion struggle to make significant progress, given the abstract nature of the open-ended exploration goal that is challenging to execute without an appropriate curriculum.

**Consistent tech tree mastery.** The Minecraft tech tree tests the agent's ability to craft and use a hierarchy of tools. Progressing through this tree (wooden tool → stone tool → iron tool → diamond tool) requires the agent to master systematic and compositional skills. Compared with baselines, VOYAGER unlocks the wooden level 15.3× faster (in terms of the prompting iterations), the stone level 8.5× faster, the iron level 6.4× faster, and VOYAGER is the only one to unlock the diamond level of the tech tree (Fig. 2 and Table. 1). This underscores the effectiveness of the automatic curriculum, which consistently presents challenges of suitable complexity to facilitate the agent's progress.

**Extensive map traversal.** VOYAGER is able to navigate distances 2.3× longer compared to baselines by traversing a variety of terrains, while the baseline agents often find themselves confined to local areas, which significantly hampers their capacity to discover new knowledge (Fig. 7).

**Efficient zero-shot generalization to unseen tasks.** To evaluate zero-shot generalization, we clear the agent's inventory, reset it to a newly instantiated world, and test it with unseen tasks. For both VOYAGER

Table 1: Tech tree mastery. Fractions indicate the number of successful trials out of three total runs. 0/3 means the method fails to unlock a level of the tech tree within the maximal prompting iterations (160). Numbers are prompting iterations averaged over three trials. The fewer the iterations, the more efficient the method.

| Method | Wooden Tool | Stone Tool | Iron Tool | Diamond Tool |
|---|---|---|---|---|
| ReAct (Yao et al., 2022) | N/A ($^0/_3$) | N/A ($^0/_3$) | N/A ($^0/_3$) | N/A ($^0/_3$) |
| Reflexion (Shinn et al., 2023) | N/A ($^0/_3$) | N/A ($^0/_3$) | N/A ($^0/_3$) | N/A ($^0/_3$) |
| AutoGPT (Richards, 2023) | $92 \pm 72$ ($^3/_3$) | $94 \pm 72$ ($^3/_3$) | $135 \pm 103$ ($^3/_3$) | N/A ($^0/_3$) |
| VOYAGER w/o Skill Library | **$7 \pm 2$ ($^3/_3$)** | **$9 \pm 4$ ($^3/_3$)** | $29 \pm 11$ ($^3/_3$) | N/A ($^0/_3$) |
| VOYAGER (Ours) | **$6 \pm 2$ ($^3/_3$)** | **$11 \pm 2$ ($^3/_3$)** | **$21 \pm 7$ ($^3/_3$)** | **$102$ ($^1/_3$)** |

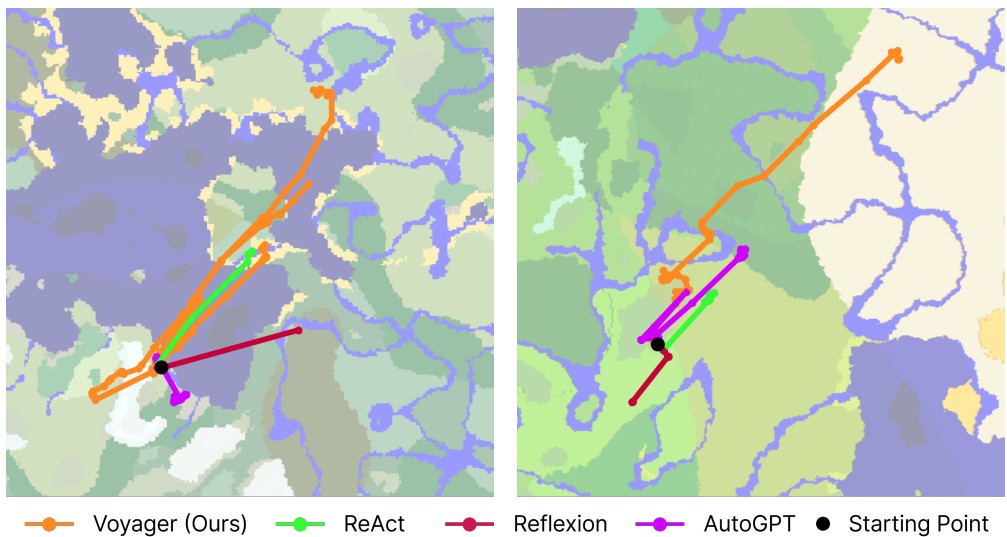

Figure 7: Map coverage: Two bird's eye views of Minecraft maps. VOYAGER is able to traverse 2.3× longer distances compared to baselines while crossing diverse terrains. Trajectories are plotted based on the positions where each agent interacts with GPT-4.

Table 2: Zero-shot generalization to unseen tasks. Fractions indicate the number of successful trials out of three total attempts. 0/3 means the method fails to solve the task within the maximal prompting iterations (50). Numbers are prompting iterations averaged over three trials. The fewer the iterations, the more efficient the method.

| Method | Diamond Pickaxe | Golden Sword | Lava Bucket | Compass |
|---|---|---|---|---|
| ReAct (Yao et al., 2022) | N/A ($^0/_3$) | N/A ($^0/_3$) | N/A ($^0/_3$) | N/A ($^0/_3$) |
| Reflexion (Shinn et al., 2023) | N/A ($^0/_3$) | N/A ($^0/_3$) | N/A ($^0/_3$) | N/A ($^0/_3$) |
| AutoGPT (Richards, 2023) | N/A ($^0/_3$) | N/A ($^0/_3$) | N/A ($^0/_3$) | N/A ($^0/_3$) |
| AutoGPT (Richards, 2023) w/ Our Skill Library | $39$ ($^1/_3$) | $30$ ($^1/_3$) | N/A ($^0/_3$) | $30$ ($^2/_3$) |
| VOYAGER w/o Skill Library | $36$ ($^2/_3$) | $30 \pm 9$ ($^3/_3$) | $27 \pm 9$ ($^3/_3$) | $26 \pm 3$ ($^3/_3$) |
| VOYAGER (Ours) | **$19 \pm 3$ ($^3/_3$)** | **$18 \pm 7$ ($^3/_3$)** | **$21 \pm 5$ ($^3/_3$)** | **$18 \pm 2$ ($^3/_3$)** |

and AutoGPT, we utilize GPT-4 to break down the task into a series of subgoals. Table. 2 and Fig. 8 show VOYAGER can consistently solve all the tasks, while baselines cannot solve any task within 50 prompting iterations. What's interesting to note is that our skill library constructed from lifelong learning not only enhances VOYAGER's performance but also gives a boost to AutoGPT. This demonstrates that the skill library serves as a versatile tool that can be readily employed by other methods, effectively acting as a plug-and-play asset to enhance performance.

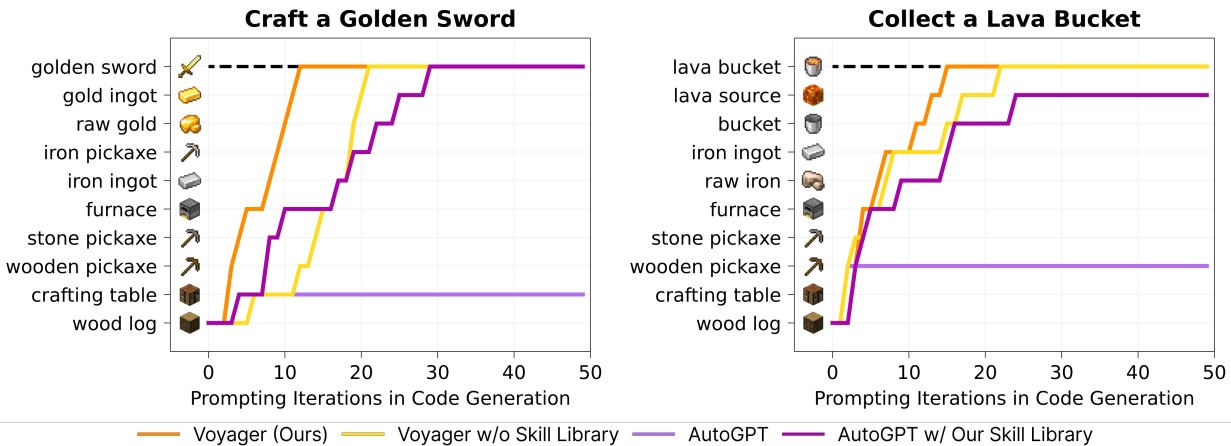

Figure 8: Zero-shot generalization to unseen tasks. We visualize the intermediate progress of each method on two tasks. See Appendix, Sec. B.4.3 for the other two tasks. We do not plot ReAct and Reflexion since they do not make any meaningful progress.

## 3.4  Ablation Studies

We ablate 6 design choices (automatic curriculum, skill library, environment feedback, execution errors, self-verification, and GPT-4 for code generation) in VOYAGER and study their impact on exploration performance (see Appendix, Sec. B.3 for details of each ablated variant). Results are shown in Fig. 9. We highlight the key findings below:

- **Automatic curriculum is crucial for the agent's consistent progress.** The discovered item count drops by 93% if the curriculum is replaced with a random one, because certain tasks may be too challenging if attempted out of order. On the other hand, a manually designed curriculum requires significant Minecraft-specific expertise, and does not take into account the agent's live situation. It falls short in the experimental results compared to our automatic curriculum.

- **Voyager w/o skill library exhibits a tendency to plateau in the later stages.** This underscores the pivotal role that the skill library plays in VOYAGER. It helps create more complex actions and steadily pushes the agent's boundaries by encouraging new skills to be built upon older ones.

- **Self-verification is the most important among all the feedback types**. Removing the module leads to a significant drop ($-73\%$) in the discovered item count. Self-verification serves as a critical mechanism to decide when to move on to a new task or reattempt a previously unsuccessful task.

- **GPT-4 significantly outperforms GPT-3.5 in code generation** and obtains $5.7\times$ more unique items, as GPT-4 exhibits a quantum leap in coding abilities. This finding corroborates recent studies in the literature  (Bubeck et al., 2023; Liu et al., 2023b).

## 4  Limitations and Future Work

**Multimodal Feedback.** VOYAGER does not currently support visual perception, because the available version of GPT-4 API is text-only at the time of this writing. However, VOYAGER has the potential to be augmented by multimodal perception models (Liu et al., 2023a; Driess et al., 2023) to achieve more impressive tasks. We demonstrate that given human feedback, VOYAGER is able to construct complex 3D structures in Minecraft, such as a Nether Portal and a house (Fig. 10). There are two ways to integrate human feedback:

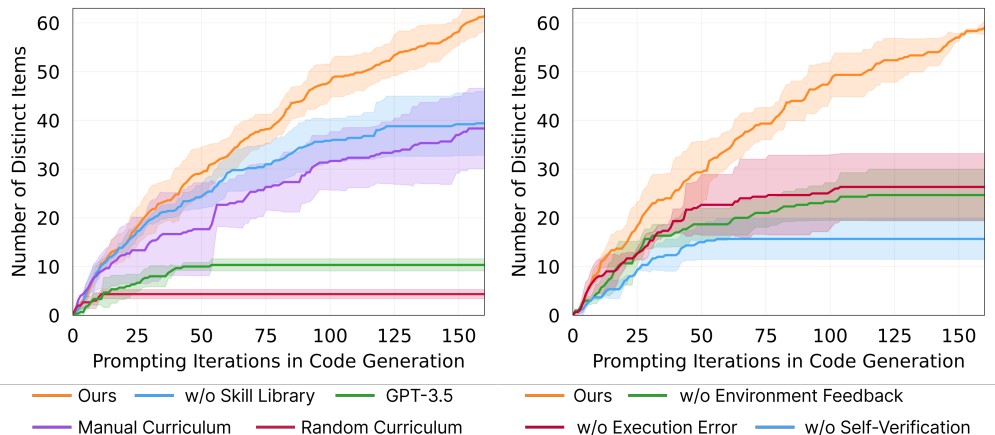

Figure 9: **Left: Ablation studies for the automatic curriculum, skill library, and GPT-4.** GPT-3.5 means replacing GPT-4 with GPT-3.5 for code generation. VOYAGER outperforms all the alternatives, demonstrating the critical role of each component. **Right: Ablation studies for the iterative prompting mechanism.** VOYAGER surpasses all the other options, thereby highlighting the essential significance of each type of feedback in the iterative prompting mechanism.

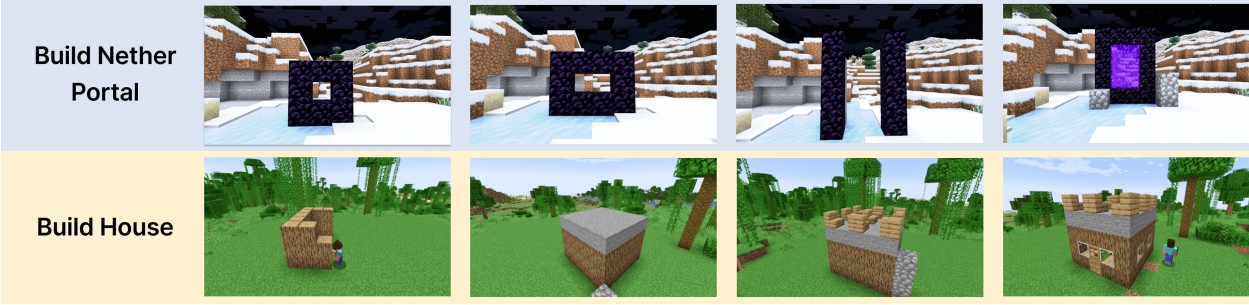

Figure 10: VOYAGER builds 3D structures with human feedback. The progress of building designs that integrate human input is demonstrated from left to right.

(1) Human as a critic (equivalent to VOYAGER's self-verification module): humans provide visual critique to VOYAGER, allowing it to modify the code from the previous round. This feedback is essential for correcting certain errors in the spatial details of a 3D structure that VOYAGER cannot perceive directly.

(2) Human as a curriculum (equivalent to VOYAGER's automatic curriculum module): humans break down a complex building task into smaller steps, guiding VOYAGER to complete them incrementally. This approach improves VOYAGER's ability to handle more sophisticated 3D construction tasks.

**Cost.** The GPT-4 API incurs significant costs. It is 15× more expensive than GPT-3.5. Nevertheless, VOYAGER requires the quantum leap in code generation quality from GPT-4 (Fig. 9), which GPT-3.5 and open-source LLMs cannot provide (Touvron et al., 2023).

**Inaccuracies.** Despite the iterative prompting mechanism, there are still cases where the agent gets stuck and fails to generate the correct skill. The automatic curriculum has the flexibility to reattempt this task at a later time. Occasionally, self-verification module may also fail, such as not recognizing spider string as a success signal of beating a spider.

**Hallucinations.** The automatic curriculum occasionally proposes unachievable tasks. For example, it may ask the agent to craft a "copper sword" or "copper chestplate", which are items that do not exist within the game. Hallucinations also occur during the code generation process. For instance, GPT-4 tends to

use cobblestone as a fuel input, despite being an invalid fuel source in the game. Additionally, it may call functions absent in the provided control primitive APIs, leading to code execution errors.

We are confident that improvements in the GPT API models as well as novel techniques for finetuning open-source LLMs will overcome these limitations in the future.

## 5 Related work

**Decision-making Agents in Minecraft.** Minecraft is an open-ended 3D world with incredibly flexible game mechanics supporting a broad spectrum of activities. Built upon notable Minecraft benchmarks (Fan et al., 2022; Guss et al., 2019b;a; 2021; Kanervisto et al., 2022; Johnson et al., 2016), Minecraft learning algorithms can be divided into two categories: 1) Low-level controller: Many prior efforts leverage hierarchical reinforcement learning to learn from human demonstrations (Lin et al., 2021; Mao et al., 2021; Skrynnik et al., 2021). Kanitscheider et al. (2021) design a curriculum based on success rates, but its objectives are limited to curated items. MineDojo (Fan et al., 2022) and VPT (Baker et al., 2022) utilize YouTube videos for large-scale pre-training. DreamerV3 (Hafner et al., 2023), on the other hand, learns a world model to explore the environment and collect diamonds. 2) High-level planner: Volum et al. (2022) leverage few-shot prompting with Codex (Chen et al., 2021a) to generate executable policies, but they require additional human interaction. Recent works leverage LLMs as a high-level planner in Minecraft by decomposing a high-level task into several subgoals following Minecraft recipes (Wang et al., 2023b; Nottingham et al., 2023; Yuan et al., 2023), thus lacking full exploration flexibility. Like these latter works, VOYAGER also uses LLMs as a high-level planner by prompting GPT-4 and utilizes Mineflayer (PrismarineJS, 2013) as a low-level controller following Volum et al. (2022). Unlike prior works, VOYAGER employs an automatic curriculum that unfolds in a bottom-up manner, driven by curiosity, and therefore enables open-ended exploration.

**Large Language Models for Agent Planning.** Inspired by the strong emergent capabilities of LLMs, such as zero-shot prompting and complex reasoning (Bommasani et al., 2021; Brown et al., 2020; Raffel et al., 2020; Wei et al., 2022a; Chowdhery et al., 2022; Chung et al., 2022), embodied agent research (Duan et al., 2022; Batra et al., 2020; Ravichandar et al., 2020; Collins et al., 2021) has witnessed a significant increase in the utilization of LLMs for planning purposes. Recent efforts can be roughly classified into two groups. 1) Large language models for robot learning: Many prior works apply LLMs to generate subgoals for robot planning (Huang et al., 2022a;a; Ahn et al., 2022; Min et al., 2021; Blukis et al., 2021). Inner Monologue (Huang et al., 2022b) incorporates environment feedback for robot planning with LLMs. Code as Policies (Liang et al., 2022) and ProgPrompt (Singh et al., 2022) directly leverage LLMs to generate executable robot policies. VIMA (Jiang et al., 2022) and PaLM-E (Driess et al., 2023) fine-tune pre-trained LLMs to support multimodal prompts. 2) Large language models for text agents: ReAct (Yao et al., 2022) leverages chain-of-thought prompting (Wei et al., 2022b) and generates both reasoning traces and task-specific actions with LLMs. Reflexion (Shinn et al., 2023) is built upon ReAct (Yao et al., 2022) with self-reflection to enhance reasoning. AutoGPT (Richards, 2023) is a popular tool that automates NLP tasks by crafting a curriculum of multiple subgoals for completing a high-level goal while incorporating ReAct (Yao et al., 2022)'s reasoning and acting loops. DERA (Nair et al., 2023) frames a task as a dialogue between two GPT-4 (OpenAI, 2023) agents. Generative Agents (Park et al., 2023) leverages ChatGPT (chatgpt) to simulate human behaviors by storing agents' experiences as memories and retrieving those for planning, but its agent actions are not executable. SPRING (Wu et al., 2023) is a concurrent work that uses GPT-4 to extract game mechanics from game manuals, based on which it answers questions arranged in a directed acyclic graph and predicts the next action. All these works lack a skill library for developing more complex behaviors, which are crucial components for the success of VOYAGER in lifelong learning.

**Code Generation with Execution.** Code generation has been a longstanding challenge in NLP (Chen et al., 2021a; Nijkamp et al., 2022; Le et al., 2022; Chowdhery et al., 2022; Brown et al., 2020), with various works leveraging execution results to improve program synthesis. Execution-guided approaches leverage intermediate execution outcomes to guide program search (Chen et al., 2019; 2021b; Ellis et al., 2019). Another line of research utilizes majority voting to choose candidates based on their execution performance (Li et al., 2022; Cobbe et al., 2021). Additionally, LEVER (Ni et al., 2023) trains a verifier to distinguish and reject incorrect programs based on execution results. CLAIRIFY (Skreta et al., 2023), on the other hand, generates code for

planning chemistry experiments and makes use of a rule-based verifier to iteratively provide error feedback to LLMs. Voyager distinguishes itself from these works by integrating environment feedback, execution errors, and self-verification (to assess task success) into an iterative prompting mechanism for embodied control.

## 6 Conclusion

In this work, we introduce Voyager, the first LLM-powered embodied lifelong learning agent, which leverages GPT-4 to explore the world continuously, develop increasingly sophisticated skills, and make new discoveries consistently without human intervention. Voyager exhibits superior performance in discovering novel items, unlocking the Minecraft tech tree, traversing diverse terrains, and applying its learned skill library to unseen tasks in a newly instantiated world. Voyager serves as a starting point to develop powerful generalist agents without tuning the model parameters.

## 7 Broader Impacts

Our research is conducted within Minecraft, a safe and harmless 3D video game environment. While Voyager is designed to be generally applicable to other domains, such as robotics, its application to physical robots would require additional attention and the implementation of safety constraints by humans to ensure responsible and secure deployment.

## 8 Acknowledgements

We are extremely grateful to Ziming Zhu, Kaiyu Yang, Rafał Kocielnik, Colin White, Or Sharir, Sahin Lale, De-An Huang, Jean Kossaifi, Yuncong Yang, Charles Zhang, Minchao Huang, and many other colleagues and friends for their helpful feedback and insightful discussions. This work is done during Guanzhi Wang's internship at NVIDIA. Guanzhi Wang is supported by the Kortschak fellowship in Computing and Mathematical Sciences at Caltech.

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

# A  Method

## A.1  Prompting

GPT-4 and GPT-3.5 offer users the ability to designate the role of each prompt message among three options:

- System: A high-level instruction that guides the model behavior throughout the conversation. It sets the overall tone and objective for the interaction.

- User: A detailed instruction that guides the assistant for the next immediate response.

- Assistant: A response message generated the model.

See `https://platform.openai.com/docs/guides/chat/introduction` for more details.

To save token usage, instead of engaging in multi-round conversations, we concatenate a system prompt and a user prompt to obtain each assistant's response.

## A.2  Automatic Curriculum

### A.2.1  Components in the Prompt

The input prompt to GPT-4 consists of several components:

(1) Directives encouraging diverse behaviors and imposing constraints (so that the proposed task is achievable and verifiable): See Sec. A.2.4 for the full prompt;

(2) The agent's current state:

- **Inventory**: A dictionary of items with counts, for example, {'cobblestone': 4, 'furnace': 1, 'stone_pickaxe': 1, 'oak_planks': 7, 'dirt': 6, 'wooden_pickaxe': 1, 'crafting_table': 1, 'raw_iron': 4, 'coal': 1};
- **Equipment**: Armors or weapons equipped by the agents;
- **Nearby blocks**: A set of block names within a 32-block distance to the agent, for example, 'dirt', 'water', 'spruce_planks', 'grass_block', 'dirt_path', 'sugar_cane', 'fern';
- **Other blocks that are recently seen**: Blocks that are not nearby or in the inventory;
- **Nearby entities**: A set of entity names within a 32-block distance to the agent, for example, 'pig', 'cat', 'villager', 'zombie';
- **A list of chests that are seen by the agent**: Chests are external containers where the agent can deposit items. If a chest is not opened before, its content is "Unknown". Otherwise, the items inside each chest are shown to the agent.
- **Biome**: For example, 'plains', 'flower_forest', 'meadow', 'river', 'beach', 'forest', 'snowy_slopes', 'frozen_peaks', 'old_growth_birch_forest', 'ocean', 'sunflower_plains', 'stony_shore';
- **Time**: One of 'sunrise', 'day', 'noon', 'sunset', 'night', 'midnight';
- **Health and hunger bars**: The max value is 20;
- **Position**: 3D coordinate $(x, y, z)$ of the agent's position in the Minecraft world;

(3) Previously completed and failed tasks;

(4) Additional context: See Sec. A.2.2;

(5) Chain-of-thought prompting (Wei et al., 2022b) in response: We request GPT-4 to first reason about the current progress and then suggest the next task.

### A.2.2 Additional Context

We leverage GPT-3.5 to self-ask questions to provide additional context. Each question is paired with a concept that is used for retrieving the most relevant document from the wiki knowledge base (Fan et al., 2022). We feed the document content to GPT-3.5 for self-answering questions. In practice, using a wiki knowledge base is optional since GPT-3.5 already possesses a good understanding of Minecraft game mechanics. However, the external knowledge base becomes advantageous if GPT-3.5 is not pre-trained in that specific domain. See Sec. A.2.4 for the full prompt.

### A.2.3 Warm-up Schedule

In practice, we adopt a warm-up schedule to gradually incorporate the agent's state and the additional context into the prompt based on how many tasks the agent has completed. This ensures that the prompt is exposed to increasing amounts of information over the exploration progress and therefore begins with basic skills and progressively advances towards more intricate and diverse ones. The warm-up setting that we use across all the experiments is shown in Table. A.1.

Table A.1: Warm-up schedule for automatic curriculum.

| Information in the prompt | After how many tasks are completed |
| --- | --- |
| core inventory (only including log, planks, stick, crafting table, furnace, dirt, coal, pickaxe, sword, and axe) | 0 |
| equipment | 0 |
| nearby blocks | 0 |
| position | 0 |
| nearby entities | 5 |
| full inventory | 7 |
| other blocks that are recently seen | 10 |
| biome | 10 |
| health bar | 15 |
| hunger bar | 15 |
| time | 15 |
| additional context | 15 |

### A.2.4 Full Prompt

Prompt 1: Full system prompt for automatic curriculum. The list of question-answer pairs represents the additional context.

```
You are a helpful assistant that tells me the next immediate task to do in Minecraft. My
    ultimate goal is to discover as many diverse things as possible, accomplish as many
    diverse tasks as possible and become the best Minecraft player in the world.

I will give you the following information:
Question 1: ...
Answer: ...
Question 2: ...
Answer: ...
Question 3: ...
Answer: ...
...
Biome: ...
Time: ...
Nearby blocks: ...
Other blocks that are recently seen: ...
Nearby entities (nearest to farthest): ...
```

```
Health: Higher than 15 means I'm healthy.
Hunger: Higher than 15 means I'm not hungry.
Position: ...
Equipment: If I have better armor in my inventory, you should ask me to equip it.
Inventory (xx/36): ...
Chests: You can ask me to deposit or take items from these chests. There also might be some
    unknown chest, you should ask me to open and check items inside the unknown chest.
Completed tasks so far: ...
Failed tasks that are too hard: ...

You must follow the following criteria:
1) You should act as a mentor and guide me to the next task based on my current learning
    progress.
2) Please be very specific about what resources I need to collect, what I need to craft, or
    what mobs I need to kill.
3) The next task should follow a concise format, such as "Mine [quantity] [block]", "Craft [
    quantity] [item]", "Smelt [quantity] [item]", "Kill [quantity] [mob]", "Cook [quantity]
    [food]", "Equip [item]" etc. It should be a single phrase. Do not propose multiple tasks
     at the same time. Do not mention anything else.
4) The next task should not be too hard since I may not have the necessary resources or have
     learned enough skills to complete it yet.
5) The next task should be novel and interesting. I should look for rare resources, upgrade
    my equipment and tools using better materials, and discover new things. I should not be
    doing the same thing over and over again.
6) I may sometimes need to repeat some tasks if I need to collect more resources to complete
     more difficult tasks. Only repeat tasks if necessary.
7) Do not ask me to build or dig shelter even if it's at night. I want to explore the world
    and discover new things. I don't want to stay in one place.
8) Tasks that require information beyond the player's status to verify should be avoided.
    For instance, "Placing 4 torches" and "Dig a 2x1x2 hole" are not ideal since they
    require visual confirmation from the screen. All the placing, building, planting, and
    trading tasks should be avoided. Do not propose task starting with these keywords.

You should only respond in the format as described below:
RESPONSE FORMAT:
Reasoning: Based on the information I listed above, do reasoning about what the next task
    should be.
Task: The next task.

Here's an example response:
Reasoning: The inventory is empty now, chop down a tree to get some wood.
Task: Obtain a wood log.
```

Prompt 2: Full system prompt for asking questions. We provide both good and bad examples as few-shot exemplars.

```
You are a helpful assistant that asks questions to help me decide the next immediate task to
    do in Minecraft. My ultimate goal is to discover as many things as possible, accomplish
    as many tasks as possible and become the best Minecraft player in the world.

I will give you the following information:
Biome: ...
Time: ...
Nearby blocks: ...
Other blocks that are recently seen: ...
Nearby entities (nearest to farthest): ...
Health: ...
Hunger: ...
Position: ...
Equipment: ...
Inventory (xx/36): ...
Chests: ...
Completed tasks so far: ...
Failed tasks that are too hard: ...

You must follow the following criteria:
```

```
1) You should ask at least 5 questions (but no more than 10 questions) to help me decide the
     next immediate task to do. Each question should be followed by the concept that the
     question is about.
2) Your question should be specific to a concept in Minecraft.
  Bad example (the question is too general):
    Question: What is the best way to play Minecraft?
    Concept: unknown
  Bad example (axe is still general, you should specify the type of axe such as wooden axe):
    What are the benefits of using an axe to gather resources?
    Concept: axe
  Good example:
    Question: How to make a wooden pickaxe?
    Concept: wooden pickaxe
3) Your questions should be self-contained and not require any context.
  Bad example (the question requires the context of my current biome):
    Question: What are the blocks that I can find in my current biome?
    Concept: unknown
  Bad example (the question requires the context of my current inventory):
    Question: What are the resources you need the most currently?
    Concept: unknown
  Bad example (the question requires the context of my current inventory):
    Question: Do you have any gold or emerald resources?
    Concept: gold
  Bad example (the question requires the context of my nearby entities):
    Question: Can you see any animals nearby that you can kill for food?
    Concept: food
  Bad example (the question requires the context of my nearby blocks):
    Question: Is there any water source nearby?
    Concept: water
  Good example:
    Question: What are the blocks that I can find in the sparse jungle?
    Concept: sparse jungle
4) Do not ask questions about building tasks (such as building a shelter) since they are too
     hard for me to do.

Let's say your current biome is sparse jungle. You can ask questions like:
Question: What are the items that I can find in the sparse jungle?
Concept: sparse jungle
Question: What are the mobs that I can find in the sparse jungle?
Concept: sparse jungle

Let's say you see a creeper nearby, and you have not defeated a creeper before. You can ask
    a question like:
Question: How to defeat the creeper?
Concept: creeper

Let's say you last completed task is "Craft a wooden pickaxe". You can ask a question like:
Question: What are the suggested tasks that I can do after crafting a wooden pickaxe?
Concept: wooden pickaxe

Here are some more question and concept examples:
Question: What are the ores that I can find in the sparse jungle?
Concept: sparse jungle
(the above concept should not be "ore" because I need to look up the page of "sparse jungle"
     to find out what ores I can find in the sparse jungle)
Question: How can you obtain food in the sparse jungle?
Concept: sparse jungle
(the above concept should not be "food" because I need to look up the page of "sparse jungle
    " to find out what food I can obtain in the sparse jungle)
Question: How can you use the furnace to upgrade your equipment and make useful items?
Concept: furnace
Question: How to obtain a diamond ore?
Concept: diamond ore
Question: What are the benefits of using a stone pickaxe over a wooden pickaxe?
Concept: stone pickaxe
Question: What are the tools that you can craft using wood planks and sticks?
Concept: wood planks
```

```
You should only respond in the format as described below:
RESPONSE FORMAT:
Reasoning: ...
Question 1: ...
Concept 1: ...
Question 2: ...
Concept 2: ...
Question 3: ...
Concept 3: ...
Question 4: ...
Concept 4: ...
Question 5: ...
Concept 5: ...
...
```

Prompt 3: Full system prompt for answering questions. Context represents the optional content from a wiki knowledge base.

```
You are a helpful assistant that answer my question about Minecraft.

I will give you the following information:
Question: ...

You will answer the question based on the context (only if available and helpful) and your
    own knowledge of Minecraft.
1) Start your answer with "Answer: ".
2) Answer "Answer: Unknown" if you don't know the answer.
```

### A.3 Skill Library

#### A.3.1 Components in the Prompt

The input prompt to GPT-4 consists of the following components:

(1) Guidelines for code generation: See Sec A.3.2 for the full prompt;

(2) Control primitive APIs implemented by us: These APIs serve a dual purpose: they demonstrate the usage of Mineflayer APIs, and they can be directly called by GPT-4.

- `exploreUntil(bot, direction, maxTime = 60, callback)`: Allow the agent to explore in a fixed direction for `maxTime`. The `callback` is the stopping condition implemented by the agent to determine when to stop exploring;

- `mineBlock(bot, name, count = 1)`: Mine and collect the specified number of blocks within a 32-block distance;

- `craftItem(bot, name, count = 1)`: Craft the item with a crafting table nearby;

- `placeItem(bot, name, position)`: Place the block at the specified position;

- `smeltItem(bot, itemName, fuelName, count = 1)`: Smelt the item with the specified fuel. There must be a furnace nearby;

- `killMob(bot, mobName, timeout = 300)`: Attack the mob and collect its dropped item;

- `getItemFromChest(bot, chestPosition, itemsToGet)`: Move to the chest at the specified position and get items from the chest;

- `depositItemIntoChest(bot, chestPosition, itemsToDeposit)`: Move to the chest at the specified position and deposit items into the chest;

(3) Control primitive APIs provided by Mineflayer:

- `await bot.pathfinder.goto(goal)`: Go to a specific position. See below for how to set the goal;

- new `GoalNear(x, y, z, range)`: Move the bot to a block within the specified range of the specified block;
- new `GoalXZ(x, z)`: For long-range goals that don't have a specific Y level;
- new `GoalGetToBlock(x, y, z)`: Not get into the block, but get directly adjacent to it. Useful for fishing, farming, filling a bucket, and using a bed.;
- new `GoalFollow(entity, range)`: Follow the specified entity within the specified range;
- new `GoalPlaceBlock(position, bot.world, {})`: Position the bot in order to place a block;
- new `GoalLookAtBlock(position, bot.world, {})`: Path towards a position where a face of the block at `position` is visible;
- `bot.isABed(bedBlock)`: Return true if `bedBlock` is a bed;
- `bot.blockAt(position)`: Return the block at `position`;
- `await bot.equip(item, destination)`: Equip the item in the specified destination. `destination` must be one of "hand", "head", "torso", "legs", "feet", "off-hand";
- `await bot.consume()`: Consume the item in the bot's hand. You must equip the item to consume first. Useful for eating food, drinking potions, etc.;
- `await bot.fish()`: Let bot fish. Before calling this function, you must first get to a water block and then equip a fishing rod. The bot will automatically stop fishing when it catches a fish;
- `await bot.sleep(bedBlock)`: Sleep until sunrise. You must get to a bed block first;
- `await bot.activateBlock(block)`: This is the same as right-clicking a block in the game. Useful for buttons, doors, etc. You must get to the block first;
- `await bot.lookAt(position)`: Look at the specified position. You must go near the position before you look at it. To fill a bucket with water, you must look at it first;
- `await bot.activateItem()`: This is the same as right-clicking to use the item in the bot's hand. Useful for using a bucket, etc. You must equip the item to activate first;
- `await bot.useOn(entity)`: This is the same as right-clicking an entity in the game. Useful for shearing a sheep. You must get to the entity first;

(4) Retrieved skills from the skill library;

(5) Generated code from the last round;

(6) Environment feedback: The chat log in the prompt;

(7) Execution errors;

(8) Critique from the self-verification module;

(9) The agent's current state: See Sec. A.2.1 for each element of the agent's state;

(10) Task proposed by the automatic curriculum;

(11) Task context: We prompt GPT-3.5 to ask for general suggestions about how to solve the task. In practice, this part is handled by the automatic curriculum since it has a systematic mechanism for question-answering (Sec. A.2.2);

(12) Chain-of-thought prompting (Wei et al., 2022b) in response: We ask GPT-4 to first explain the reason why the code from the last round fails, then give step-by-step plans to finish the task, and finally generate code. See Sec. A.3.2 for the full prompt.

### A.3.2 Full Prompt

Prompt 4: Full system prompt for code generation.

```
You are a helpful assistant that writes Mineflayer javascript code to complete any Minecraft
    task specified by me.

Here are some useful programs written with Mineflayer APIs.

/*
Explore until find an iron_ore, use Vec3(0, -1, 0) because iron ores are usually underground
await exploreUntil(bot, new Vec3(0, -1, 0), 60, () => {
    const iron_ore = bot.findBlock({
        matching: mcData.blocksByName["iron_ore"].id,
        maxDistance: 32,
    });
    return iron_ore;
});

Explore until find a pig, use Vec3(1, 0, 1) because pigs are usually on the surface
let pig = await exploreUntil(bot, new Vec3(1, 0, 1), 60, () => {
    const pig = bot.nearestEntity((entity) => {
        return (
            entity.name === "pig" &&
            entity.position.distanceTo(bot.entity.position) < 32
        );
    });
    return pig;
});
*/
async function exploreUntil(bot, direction, maxTime = 60, callback) {
    /*
    Implementation of this function is omitted.
    direction: Vec3, can only contain value of -1, 0 or 1
    maxTime: number, the max time for exploration
    callback: function, early stop condition, will be called each second, exploration will
    stop if return value is not null

    Return: null if explore timeout, otherwise return the return value of callback
    */
}

// Mine 3 cobblestone: mineBlock(bot, "stone", 3);
async function mineBlock(bot, name, count = 1) {
    const blocks = bot.findBlocks({
        matching: (block) => {
            return block.name === name;
        },
        maxDistance: 32,
        count: count,
    });
    const targets = [];
    for (let i = 0; i < Math.min(blocks.length, count); i++) {
        targets.push(bot.blockAt(blocks[i]));
    }
    await bot.collectBlock.collect(targets, { ignoreNoPath: true });
}

// Craft 8 oak_planks from 2 oak_log (do the recipe 2 times): craftItem(bot, "oak_planks",
    2);
// You must place a crafting table before calling this function
async function craftItem(bot, name, count = 1) {
    const item = mcData.itemsByName[name];
    const craftingTable = bot.findBlock({
        matching: mcData.blocksByName.crafting_table.id,
        maxDistance: 32,
```

```
    });
    await bot.pathfinder.goto(
        new GoalLookAtBlock(craftingTable.position, bot.world)
    );
    const recipe = bot.recipesFor(item.id, null, 1, craftingTable)[0];
    await bot.craft(recipe, count, craftingTable);
}

// Place a crafting_table near the player, Vec3(1, 0, 0) is just an example, you shouldn't
    always use that: placeItem(bot, "crafting_table", bot.entity.position.offset(1, 0, 0));
async function placeItem(bot, name, position) {
    const item = bot.inventory.findInventoryItem(mcData.itemsByName[name].id);
    // find a reference block
    const faceVectors = [
        new Vec3(0, 1, 0),
        new Vec3(0, -1, 0),
        new Vec3(1, 0, 0),
        new Vec3(-1, 0, 0),
        new Vec3(0, 0, 1),
        new Vec3(0, 0, -1),
    ];
    let referenceBlock = null;
    let faceVector = null;
    for (const vector of faceVectors) {
        const block = bot.blockAt(position.minus(vector));
        if (block?.name !== "air") {
            referenceBlock = block;
            faceVector = vector;
            break;
        }
    }
    // You must first go to the block position you want to place
    await bot.pathfinder.goto(new GoalPlaceBlock(position, bot.world, {}));
    // You must equip the item right before calling placeBlock
    await bot.equip(item, "hand");
    await bot.placeBlock(referenceBlock, faceVector);
}

// Smelt 1 raw_iron into 1 iron_ingot using 1 oak_planks as fuel: smeltItem(bot, "raw_iron",
     "oak_planks");
// You must place a furnace before calling this function
async function smeltItem(bot, itemName, fuelName, count = 1) {
    const item = mcData.itemsByName[itemName];
    const fuel = mcData.itemsByName[fuelName];
    const furnaceBlock = bot.findBlock({
        matching: mcData.blocksByName.furnace.id,
        maxDistance: 32,
    });
    await bot.pathfinder.goto(
        new GoalLookAtBlock(furnaceBlock.position, bot.world)
    );
    const furnace = await bot.openFurnace(furnaceBlock);
    for (let i = 0; i < count; i++) {
        await furnace.putFuel(fuel.id, null, 1);
        await furnace.putInput(item.id, null, 1);
        // Wait 12 seconds for the furnace to smelt the item
        await bot.waitForTicks(12 * 20);
        await furnace.takeOutput();
    }
    await furnace.close();
}

// Kill a pig and collect the dropped item: killMob(bot, "pig", 300);
async function killMob(bot, mobName, timeout = 300) {
    const entity = bot.nearestEntity(
```

```
        (entity) =>
            entity.name === mobName &&
            entity.position.distanceTo(bot.entity.position) < 32
    );
    await bot.pvp.attack(entity);
    await bot.pathfinder.goto(
        new GoalBlock(entity.position.x, entity.position.y, entity.position.z)
    );
}

// Get a torch from chest at (30, 65, 100): getItemFromChest(bot, new Vec3(30, 65, 100), {"
    torch": 1});
// This function will work no matter how far the bot is from the chest.
async function getItemFromChest(bot, chestPosition, itemsToGet) {
    await moveToChest(bot, chestPosition);
    const chestBlock = bot.blockAt(chestPosition);
    const chest = await bot.openContainer(chestBlock);
    for (const name in itemsToGet) {
        const itemByName = mcData.itemsByName[name];
        const item = chest.findContainerItem(itemByName.id);
        await chest.withdraw(item.type, null, itemsToGet[name]);
    }
    await closeChest(bot, chestBlock);
}
// Deposit a torch into chest at (30, 65, 100): depositItemIntoChest(bot, new Vec3(30, 65,
    100), {"torch": 1});
// This function will work no matter how far the bot is from the chest.
async function depositItemIntoChest(bot, chestPosition, itemsToDeposit) {
    await moveToChest(bot, chestPosition);
    const chestBlock = bot.blockAt(chestPosition);
    const chest = await bot.openContainer(chestBlock);
    for (const name in itemsToDeposit) {
        const itemByName = mcData.itemsByName[name];
        const item = bot.inventory.findInventoryItem(itemByName.id);
        await chest.deposit(item.type, null, itemsToDeposit[name]);
    }
    await closeChest(bot, chestBlock);
}
// Check the items inside the chest at (30, 65, 100): checkItemInsideChest(bot, new Vec3(30,
     65, 100));
// You only need to call this function once without any action to finish task of checking
    items inside the chest.
async function checkItemInsideChest(bot, chestPosition) {
    await moveToChest(bot, chestPosition);
    const chestBlock = bot.blockAt(chestPosition);
    await bot.openContainer(chestBlock);
    // You must close the chest after opening it if you are asked to open a chest
    await closeChest(bot, chestBlock);
}

await bot.pathfinder.goto(goal); // A very useful function. This function may change your
    main-hand equipment.
// Following are some Goals you can use:
new GoalNear(x, y, z, range); // Move the bot to a block within the specified range of the
    specified block. 'x', 'y', 'z', and 'range' are 'number'
new GoalXZ(x, z); // Useful for long-range goals that don't have a specific Y level. 'x' and
     'z' are 'number'
new GoalGetToBlock(x, y, z); // Not get into the block, but get directly adjacent to it.
    Useful for fishing, farming, filling bucket, and beds. 'x', 'y', and 'z' are 'number'
new GoalFollow(entity, range); // Follow the specified entity within the specified range. '
    entity' is 'Entity', 'range' is 'number'
new GoalPlaceBlock(position, bot.world, {}); // Position the bot in order to place a block.
    'position' is 'Vec3'
new GoalLookAtBlock(position, bot.world, {}); // Path into a position where a blockface of
    the block at position is visible. 'position' is 'Vec3'
```

```
// These are other Mineflayer functions you can use:
bot.isABed(bedBlock); // Return true if 'bedBlock' is a bed
bot.blockAt(position); // Return the block at 'position'. 'position' is 'Vec3'

// These are other Mineflayer async functions you can use:
await bot.equip(item, destination); // Equip the item in the specified destination. 'item'
    is 'Item', 'destination' can only be "hand", "head", "torso", "legs", "feet", "off-hand"
await bot.consume(); // Consume the item in the bot's hand. You must equip the item to
    consume first. Useful for eating food, drinking potions, etc.
await bot.fish(); // Let bot fish. Before calling this function, you must first get to a
    water block and then equip a fishing rod. The bot will automatically stop fishing when
    it catches a fish
await bot.sleep(bedBlock); // Sleep until sunrise. You must get to a bed block first
await bot.activateBlock(block); // This is the same as right-clicking a block in the game.
    Useful for buttons, doors, using hoes, etc. You must get to the block first
await bot.lookAt(position); // Look at the specified position. You must go near the position
     before you look at it. To fill bucket with water, you must lookAt first. 'position' is
     'Vec3'
await bot.activateItem(); // This is the same as right-clicking to use the item in the bot's
     hand. Useful for using buckets, etc. You must equip the item to activate first
await bot.useOn(entity); // This is the same as right-clicking an entity in the game. Useful
     for shearing sheep, equipping harnesses, etc. You must get to the entity first

{retrieved_skills}

At each round of conversation, I will give you
Code from the last round: ...
Execution error: ...
Chat log: ...
Biome: ...
Time: ...
Nearby blocks: ...
Nearby entities (nearest to farthest):
Health: ...
Hunger: ...
Position: ...
Equipment: ...
Inventory (xx/36): ...
Chests: ...
Task: ...
Context: ...
Critique: ...

You should then respond to me with
Explain (if applicable): Are there any steps missing in your plan? Why does the code not
    complete the task? What does the chat log and execution error imply?
Plan: How to complete the task step by step. You should pay attention to Inventory since it
    tells what you have. The task completeness check is also based on your final inventory.
Code:
    1) Write an async function taking the bot as the only argument.
    2) Reuse the above useful programs as much as possible.
        - Use 'mineBlock(bot, name, count)' to collect blocks. Do not use 'bot.dig' directly
    .
        - Use 'craftItem(bot, name, count)' to craft items. Do not use 'bot.craft' directly.
        - Use 'smeltItem(bot, name count)' to smelt items. Do not use 'bot.openFurnace'
    directly.
        - Use 'placeItem(bot, name, position)' to place blocks. Do not use 'bot.placeBlock'
    directly.
        - Use 'killMob(bot, name, timeout)' to kill mobs. Do not use 'bot.attack' directly.
    3) Your function will be reused for building more complex functions. Therefore, you
    should make it generic and reusable. You should not make strong assumption about the
    inventory (as it may be changed at a later time), and therefore you should always check
    whether you have the required items before using them. If not, you should first collect
    the required items and reuse the above useful programs.
    4) Functions in the "Code from the last round" section will not be saved or executed. Do
     not reuse functions listed there.
```

```
    5) Anything defined outside a function will be ignored, define all your variables inside
     your functions.
    6) Call 'bot.chat' to show the intermediate progress.
    7) Use 'exploreUntil(bot, direction, maxDistance, callback)' when you cannot find
    something. You should frequently call this before mining blocks or killing mobs. You
    should select a direction at random every time instead of constantly using (1, 0, 1).
    8) 'maxDistance' should always be 32 for 'bot.findBlocks' and 'bot.findBlock'. Do not
    cheat.
    9) Do not write infinite loops or recursive functions.
    10) Do not use 'bot.on' or 'bot.once' to register event listeners. You definitely do not
     need them.
    11) Name your function in a meaningful way (can infer the task from the name).

You should only respond in the format as described below:
RESPONSE FORMAT:
Explain: ...
Plan:
1) ...
2) ...
3) ...
...
Code:
'''javascript
// helper functions (only if needed, try to avoid them)
...
// main function after the helper functions
async function yourMainFunctionName(bot) {
  // ...
}
'''
```

Prompt 5: Full system prompt for generating function descriptions. This is used when adding a new skill to the skill library. We give a one-shot example in the prompt.

```
You are a helpful assistant that writes a description of the given function written in
    Mineflayer javascript code.

1) Do not mention the function name.
2) Do not mention anything about 'bot.chat' or helper functions.
3) There might be some helper functions before the main function, but you only need to
    describe the main function.
4) Try to summarize the function in no more than 6 sentences.
5) Your response should be a single line of text.

For example, if the function is:

async function mineCobblestone(bot) {
  // Check if the wooden pickaxe is in the inventory, if not, craft one
  let woodenPickaxe = bot.inventory.findInventoryItem(mcData.itemsByName["wooden_pickaxe"].
    id);
  if (!woodenPickaxe) {
    bot.chat("Crafting a wooden pickaxe.");
    await craftWoodenPickaxe(bot);
    woodenPickaxe = bot.inventory.findInventoryItem(mcData.itemsByName["wooden_pickaxe"].id)
    ;
  }

  // Equip the wooden pickaxe if it exists
  if (woodenPickaxe) {
    await bot.equip(woodenPickaxe, "hand");

    // Explore until we find a stone block
    await exploreUntil(bot, new Vec3(1, -1, 1), 60, () => {
      const stone = bot.findBlock({
        matching: mcData.blocksByName["stone"].id,
        maxDistance: 32
      });
```

```
        if (stone) {
          return true;
        }
    });

    // Mine 8 cobblestone blocks using the wooden pickaxe
    bot.chat("Found a stone block. Mining 8 cobblestone blocks.");
    await mineBlock(bot, "stone", 8);
    bot.chat("Successfully mined 8 cobblestone blocks.");

    // Save the event of mining 8 cobblestone
    bot.save("cobblestone_mined");
  } else {
    bot.chat("Failed to craft a wooden pickaxe. Cannot mine cobblestone.");
  }
}

The main function is 'mineCobblestone'.

Then you would write:

The function is about mining 8 cobblestones using a wooden pickaxe. First check if a wooden
    pickaxe is in the inventory. If not, craft one. If the wooden pickaxe is available,
    equip the wooden pickaxe in the hand. Next, explore the environment until finding a
    stone block. Once a stone block is found, mine a total of 8 cobblestone blocks using the
     wooden pickaxe.
```

### A.3.3 Examples

Skill library example 1: craftWoodenPlanks.

```
async function craftWoodenPlanks(bot) {
  const logNames = ["oak_log", "birch_log", "spruce_log", "jungle_log", "acacia_log", "
    dark_oak_log", "mangrove_log"];
  const plankNames = ["oak_planks", "birch_planks", "spruce_planks", "jungle_planks", "
    acacia_planks", "dark_oak_planks", "mangrove_planks"];
  const logInInventory = logNames.find(logName => bot.inventory.count(mcData.itemsByName[
    logName].id) > 0);
  if (!logInInventory) {
    bot.chat("No wooden log in inventory. Mining a wooden log...");
    await mineWoodLog(bot);
  }
  const logIndex = logNames.indexOf(logInInventory);
  const plankName = plankNames[logIndex];
  bot.chat('Crafting 4 ${plankName}...');
  await craftItem(bot, plankName, 1);
  bot.chat('4 ${plankName} crafted.');
}
```

Skill library example 2: mineTenCobbledDeepslateBelowY0.

```
async function mineTenCobbledDeepslateBelowY0(bot) {
  // Equip the iron pickaxe
  const ironPickaxe = bot.inventory.findInventoryItem(mcData.itemsByName["iron_pickaxe"].id)
    ;
  await bot.equip(ironPickaxe, "hand");

  // Find cobbled_deepslate blocks below Y=0
  const cobbledDeepslateBlocks = await exploreUntil(bot, new Vec3(1, -1, 1), 60, () => {
    const cobbledDeepslate = bot.findBlock({
      matching: mcData.blocksByName["cobbled_deepslate"].id,
      maxDistance: 32,
      position: pos => pos.y < 0
    });
    return cobbledDeepslate;
  });
```

```
  // Mine 10 cobbled_deepslate blocks
  await mineBlock(bot, "cobbled_deepslate", 10);
  bot.chat("10 cobbled_deepslate mined below Y=0.");
}
```

Skill library example 3: smeltFiveRawIronV2.

```
async function findSuitablePosition(bot) {
  const offsets = [new Vec3(1, 0, 0), new Vec3(-1, 0, 0), new Vec3(0, 0, 1), new Vec3(0, 0,
    -1)];
  for (const offset of offsets) {
    const position = bot.entity.position.offset(offset.x, offset.y, offset.z);
    const block = bot.blockAt(position);
    if (block.name === "air") {
      return position;
    }
  }
  return null;
}

async function smeltFiveRawIron(bot) {
  // Check if there is a furnace in the inventory
  const furnaceItem = bot.inventory.findInventoryItem(mcData.itemsByName.furnace.id);

  // If not, craft a furnace using the available cobblestone
  if (!furnaceItem) {
    await craftFurnace(bot);
  }

  // Find a suitable position to place the furnace
  const furnacePosition = await findSuitablePosition(bot);
  if (!furnacePosition) {
    bot.chat("Could not find a suitable position to place the furnace.");
    return;
  }

  // Place the furnace at the suitable position
  await placeItem(bot, "furnace", furnacePosition);

  // Smelt 5 raw iron using the available coal as fuel
  await smeltItem(bot, "raw_iron", "coal", 5);
  bot.chat("5 raw iron smelted.");
}
```

Skill library example 4: fillBucketWithWater.

```
async function fillBucketWithWater(bot) {
// Find a water block nearby
  const waterBlock = await exploreUntil(bot, new Vec3(1, 0, 1), 60, () => {
    const water = bot.findBlock({
      matching: mcData.blocksByName.water.id,
      maxDistance: 32
    });
    return water;
  });
  if (!waterBlock) {
    bot.chat("Could not find water.");
    return;
  }

  // Go to the water block
  const adjacentBlock= waterBlock.position.offset(0, 1, 0);
  await bot.pathfinder.goto(
      new GoalGetToBlock(adjacentBlock.x, adjacentBlock.y, adjacentBlock.z)
  );
```

```
    // Look at the water block
    await bot.lookAt(waterBlock.position, true);

    // Equip the bucket
    const bucket = bot.inventory.findInventoryItem(mcData.itemsByName.bucket.id);
    await bot.equip(bucket, "hand");

    // Activate the bucket to collect water
    await bot.activateItem();
    bot.chat("Filled the bucket with water.");
}
```

Skill library example 5: catchFiveFishSafely.

```
async function catchFiveFishSafely(bot) {
  // Check if the bot has a fishing rod in its inventory
  let fishingRod = bot.inventory.findInventoryItem(mcData.itemsByName.fishing_rod.id);
  if (!fishingRod) {
    await craftFishingRod(bot);
    fishingRod = bot.inventory.findInventoryItem(mcData.itemsByName.fishing_rod.id);
  }

  // Find a nearby water block
  let waterBlock;
  while (!waterBlock) {
    waterBlock = await exploreUntil(bot, new Vec3(1, 0, 1), 60, () => {
      const foundWaterBlock = bot.findBlock({
        matching: mcData.blocksByName.water.id,
        maxDistance: 32
      });
      return foundWaterBlock;
    });
    if (!waterBlock) {
      bot.chat("No path to the water block. Trying to find another water block...");
    }
  }

  // Move to a block adjacent to the water block
  const adjacentBlock = waterBlock.position.offset(0, 1, 0);
  await bot.pathfinder.goto(new GoalBlock(adjacentBlock.x, adjacentBlock.y, adjacentBlock.z)
    );

  // Look at the water block
  await bot.lookAt(waterBlock.position);

  // Equip the fishing rod
  await bot.equip(fishingRod, "hand");

  // Fish in the water 5 times
  for (let i = 0; i < 5; i++) {
    try {
      await bot.fish();
      bot.chat(`Fish ${i + 1} caught.`);
    } catch (error) {
      if (error.message === "Fishing cancelled") {
        bot.chat("Fishing was cancelled. Trying again...");
        i--; // Retry the same iteration
      } else {
        throw error;
      }
    }
  }
}
```

## A.4  Self-Verification

### A.4.1  Components in the Prompt

The input prompt to GPT-4 consists of the following components:

(1) The agent's state: We exclude other blocks that are recently seen and nearby entities from the agent's state since they are not useful for assessing the task's completeness. See Sec. A.2.1 for each element of the agent's state;

(2) Task proposed by the automatic curriculum;

(3) Task context: We prompt GPT-3.5 to ask for general suggestions about how to solve the task. In practice, this part is handled by the automatic curriculum since it has a systematic mechanism for question-answering (Sec. A.2.2);

(4) Chain-of-thought prompting (Wei et al., 2022b) in response: We request GPT-4 to initially reason about the task's success or failure, then output a boolean variable indicating the task's outcome, and finally provide a critique to the agent if the task fails.

(5) Few-shot examples for in-context learning (Wei et al., 2022a; Brown et al., 2020; Raffel et al., 2020).

### A.4.2  Full Prompt

Prompt 6: Full system prompt for self-verification.

```
You are an assistant that assesses my progress of playing Minecraft and provides useful
    guidance.

You are required to evaluate if I have met the task requirements. Exceeding the task
    requirements is also considered a success while failing to meet them requires you to
    provide critique to help me improve.

I will give you the following information:

Biome: The biome after the task execution.
Time: The current time.
Nearby blocks: The surrounding blocks. These blocks are not collected yet. However, this is
    useful for some placing or planting tasks.
Health: My current health.
Hunger: My current hunger level. For eating task, if my hunger level is 20.0, then I
    successfully ate the food.
Position: My current position.
Equipment: My final equipment. For crafting tasks, I sometimes equip the crafted item.
Inventory (xx/36): My final inventory. For mining and smelting tasks, you only need to check
    inventory.
Chests: If the task requires me to place items in a chest, you can find chest information
    here.
Task: The objective I need to accomplish.
Context: The context of the task.

You should only respond in JSON format as described below:
{
    "reasoning": "reasoning",
    "success": boolean,
    "critique": "critique",
}
Ensure the response can be parsed by Python 'json.loads', e.g.: no trailing commas, no
    single quotes, etc.

Here are some examples:
INPUT:
Inventory (2/36): {'oak_log':2, 'spruce_log':2}

Task: Mine 3 wood logs
```

```
RESPONSE:
{
    "reasoning": "You need to mine 3 wood logs. You have 2 oak logs and 2 spruce logs, which
     add up to 4 wood logs.",
    "success": true,
    "critique": ""
}

INPUT:
Inventory (3/36): {'crafting_table': 1, 'spruce_planks': 6, 'stick': 4}

Task: Craft a wooden pickaxe

RESPONSE:
{
    "reasoning": "You have enough materials to craft a wooden pickaxe, but you didn't craft
    it.",
    "success": false,
    "critique": "Craft a wooden pickaxe with a crafting table using 3 spruce planks and 2
    sticks."
}

INPUT:
Inventory (2/36): {'raw_iron': 5, 'stone_pickaxe': 1}

Task: Mine 5 iron_ore

RESPONSE:
{
    "reasoning": "Mining iron_ore in Minecraft will get raw_iron. You have 5 raw_iron in
    your inventory.",
    "success": true,
    "critique": ""
}

INPUT:
Biome: plains

Nearby blocks: stone, dirt, grass_block, grass, farmland, wheat

Inventory (26/36): ...

Task:  Plant 1 wheat seed.

RESPONSE:
{
    "reasoning": "For planting tasks, inventory information is useless. In nearby blocks,
    there is farmland and wheat, which means you succeed to plant the wheat seed.",
    "success": true,
    "critique": ""
}

INPUT:
Inventory (11/36): {... ,'rotten_flesh': 1}

Task: Kill 1 zombie

Context: ...

RESPONSE
{
    "reasoning": "You have rotten flesh in your inventory, which means you successfully
    killed one zombie.",
    "success": true,
    "critique": ""
}
```

```
INPUT:
Hunger: 20.0/20.0

Inventory (11/36): ...

Task: Eat 1 ...

Context: ...

RESPONSE
{
    "reasoning": "For all eating task, if the player's hunger is 20.0, then the player
    successfully ate the food.",
    "success": true,
    "critique": ""
}

INPUT:
Nearby blocks: chest

Inventory (28/36): {'rail': 1, 'coal': 2, 'oak_planks': 13, 'copper_block': 1, 'diorite': 7,
     'cooked_beef': 4, 'granite': 22, 'cobbled_deepslate': 23, 'feather': 4, 'leather': 2, '
    cooked_chicken': 3, 'white_wool': 2, 'stick': 3, 'black_wool': 1, 'stone_sword': 2, '
    stone_hoe': 1, 'stone_axe': 2, 'stone_shovel': 2, 'cooked_mutton': 4, 'cobblestone_wall
    ': 18, 'crafting_table': 1, 'furnace': 1, 'iron_pickaxe': 1, 'stone_pickaxe': 1, '
    raw_copper': 12}

Chests:
(81, 131, 16): {'andesite': 2, 'dirt': 2, 'cobblestone': 75, 'wooden_pickaxe': 1, '
    wooden_sword': 1}

Task: Deposit useless items into the chest at (81, 131, 16)

Context: ...

RESPONSE
{
    "reasoning": "You have 28 items in your inventory after depositing, which is more than
    20. You need to deposit more items from your inventory to the chest.",
    "success": false,
    "critique": "Deposit more useless items such as copper_block, diorite, granite,
    cobbled_deepslate, feather, and leather to meet the requirement of having only 20
    occupied slots in your inventory."
}
```

### A.5 System-level Comparison between Voyager and Prior Works

We make a system-level comparison in Table. A.2. Voyager stands out as the only method featuring a combination of automatic curriculum, iterative planning, and a skill library. Moreover, it learns to play Minecraft without the need for any gradient update.

## B  Experiments

### B.1 Experimental Setup

Our simulation environment is built upon MineDojo (Fan et al., 2022) and utilizes Mineflayer (PrismarineJS, 2013) JavaScript APIs for motor controls (Sec. A.3.2). Additionally, we incorporate many `bot.chat()` into Mineflayer functions to provide abundant environment feedback and implement various condition checks along with try-catch exceptions for continuous execution. If the bot dies, it is resurrected near the closest ground, and its inventory is preserved for uninterrupted exploration. The bot recycles its crafting table and furnace after program execution. For detailed implementations, please refer to our codebase.

Table A.2: System-level comparison between VOYAGER and prior works.

| | VPT (Baker et al., 2022) | DreamerV3 (Hafner et al., 2023) | DECKARD (Nottingham et al., 2023) | DEPS (Wang et al., 2023b) | Plan4MC (Yuan et al., 2023) | VOYAGER |
|---|---|---|---|---|---|---|
| Demos | Videos | None | Videos | None | None | None |
| Rewards | Sparse | Dense | Sparse | None | Dense | None |
| Observations | Pixels Only | Pixels & Meta | Pixels & Inventory | Feedback & Inventory | Pixels & Meta | Feedback & Meta & Inventory |
| Actions | Keyboard & Mouse | Discrete | Keyboard & Mouse | Keyboard & Mouse | Discrete | Code |
| Automatic Curriculum | | | ✓ | | | ✓ (in-context GPT-4 proposal) |
| Iterative Planning | | | | ✓ | | ✓ (3 types of feedback) |
| Skill Library | | | | | ✓ (pre-defined) | ✓ (self-generated) |
| Gradient-Free | | | | | | ✓ |

## B.2 Baselines

**ReAct** (Yao et al., 2022) uses chain-of-thought prompting (Wei et al., 2022b) by generating both reasoning traces and action plans with LLMs. We provide it with our environment feedback and the agent states as observations. ReAct undergoes one round of code generation from scratch, followed by three rounds of code refinement. This process is then repeated until the maximum prompting iteration is reached.

**Reflexion** (Shinn et al., 2023) is built on top of ReAct (Yao et al., 2022) with self-reflection to infer more intuitive future actions. We provide it with environment feedback, the agent states, execution errors, and our self-verification module. Similar to ReAct, Reflexion undergoes one round of code generation from scratch, followed by three rounds of code refinement. This process is then repeated until the maximum prompting iteration is reached.

**AutoGPT** (Richards, 2023) is a popular software tool that automates NLP tasks by decomposing a high-level goal into multiple subgoals and executing them in a ReAct-style loop. We re-implement AutoGPT by using GPT-4 to do task decomposition and provide it with the agent states, environment feedback, and execution errors as observations for subgoal execution. Compared with VOYAGER, AutoGPT lacks the skill library for accumulating knowledge, self-verification for assessing task success, and automatic curriculum for open-ended exploration. During each subgoal execution, if no execution error occurs, we consider the subgoal completed and proceed to the next one. Otherwise, we refine the program until three rounds of code refinement (equivalent to four rounds of code generation) are completed, and then move on to the next subgoal. If three consecutive subgoals do not result in acquiring a new item, we replan by rerunning the task decomposition.

The task is "explore the world and get as many items as possible" for all baselines.

Table A.3: Comparison between Voyager and baselines.

| | ReAct (Yao et al., 2022) | Reflexion (Shinn et al., 2023) | AutoGPT (Richards, 2023) | Voyager |
|---|---|---|---|---|
| Chain-of-Thought (Wei et al., 2022b) | ✓ | ✓ | ✓ | ✓ |
| Self Verification | | ✓ | | ✓ |
| Environment Feedback | ✓ | ✓ | ✓ | ✓ |
| Execution Errors | | ✓ | ✓ | ✓ |
| Agent State | ✓ | ✓ | ✓ | ✓ |
| Skill Library | | | | ✓ |
| Automatic Curriculum | | | | ✓ |

## B.3 Ablations

We ablate 6 design choices (automatic curriculum, skill library, environment feedback, execution errors, self-verification, and GPT-4 for code generation) in Voyager and study their impact on exploration performance.

- **Manual Curriculum**: We substitute the automatic curriculum with a manually designed curriculum: "Mine 3 wood log", "Craft 1 crafting table", "Craft 1 wooden pickaxe", "Mine 11 cobblestone", "Craft 1 stone pickaxe", "Craft 1 furnace", "Mine 3 iron ore", "Smelt 3 iron ore", "Craft 1 iron pickaxe", "Craft 1 iron sword", "Kill 3 pig", "Cook 3 porkchop", "Eat 1 cooked porkchop", "Mine 3 copper ore", "Smelt 3 copper ore", "Mine 1 amethyst", "Craft 1 spyglass", "Mine 3 gold ore", "Smelt 3 gold ore", "Craft 1 gold pickaxe", "Kill 3 cow", "Cook 3 beef", "Eat 1 cooked beef", "Mine 3 diamond ore", "Craft 1 diamond pickaxe", "Kill 3 sheep", "Cook 3 mutton", "Eat 1 cooked mutton", "Mine 1 redstone ore", "Craft 1 clock", "Craft 1 shield", "Craft 1 iron helmet", "Craft 1 iron chestplate", "Craft 1 iron leggings", "Craft 1 iron boots", "Kill 3 spider", "Craft 1 fishing rod", "Fish 3 raw cod", "Fish 3 raw salmon", "Fish 3 pufferfish", "Cook 3 raw cod", "Cook 3 raw salmon", "Craft 1 bucket", "Collect 1 water", "Mine 1 obsidian". A manual curriculum requires human effort to design and is not scalable for open-ended exploration.

- **Random Curriculum**: We curate 101 items obtained by Voyager and create a random curriculum by randomly selecting one item as the next task.

- **w/o Skill Library**: We remove the skill library, eliminating skill retrieval for code generation.

- **w/o Environment Feedback**: We exclude environment feedback (chat log) from the prompt for code generation.

- **w/o Execution Errors**: We exclude execution errors from the prompt for code generation.

- **w/o Self-Verification**: For each task, we generate code without self-verification and iteratively refine the program for 3 rounds (equivalent to 4 rounds of code generation in total).

- **GPT-3.5**: We replace GPT-4 with GPT-3.5 for code generation. We retain GPT-4 for the automatic curriculum and the self-verification module.

## B.4 Evaluation Results

### B.4.1 Significantly Better Exploration

The meaning of each icon in Fig. 1 is shown in Fig. A.1.

We run three trials for each method. The items collected by Voyager in each trial is

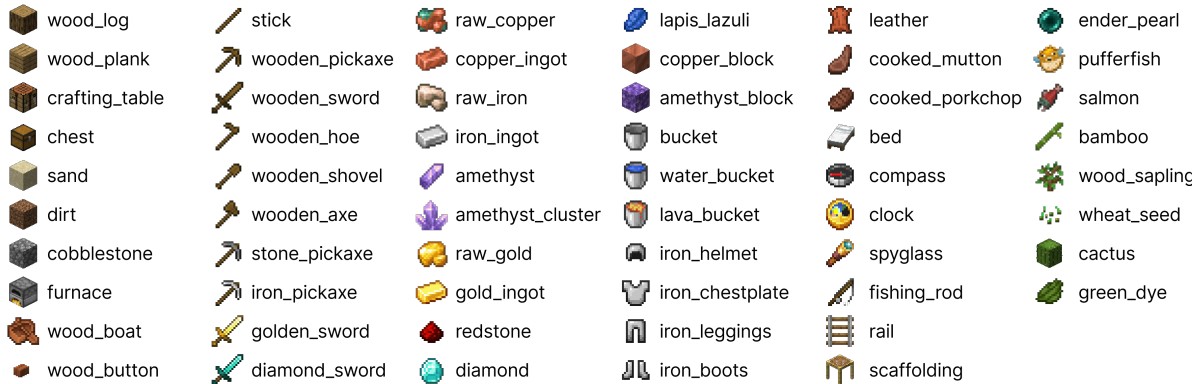

Figure A.1: Minecraft item icons with corresponding names.

- **Trial 1**: 'iron_ingot', 'stone_shovel', 'iron_leggings', 'fishing_rod', 'pufferfish', 'oak_log', 'cooked_mutton', 'green_dye', 'flint', 'chest', 'iron_sword', 'string', 'ender_pearl', 'raw_copper', 'crafting_table', 'cactus', 'lapis_lazuli', 'iron_pickaxe', 'copper_ingot', 'stone_pickaxe', 'wooden_hoe', 'scaffolding', 'stick', 'porkchop', 'copper_block', 'gravel', 'grass_block', 'white_bed', 'bone', 'dirt', 'mutton', 'white_wool', 'oak_sapling', 'coal', 'bamboo', 'wooden_pickaxe', 'rotten_flesh', 'cooked_porkchop', 'cod', 'iron_boots', 'lightning_rod', 'diorite', 'water_bucket', 'shears', 'furnace', 'andesite', 'granite', 'bucket', 'wooden_sword', 'sandstone', 'iron_helmet', 'raw_iron', 'sand', 'acacia_log', 'cooked_cod', 'oak_planks', 'azure_bluet', 'iron_shovel', 'acacia_planks', 'shield', 'iron_axe', 'iron_chestplate', 'cobblestone';

- **Trial 2**: 'iron_ingot', 'tuff', 'stone_shovel', 'iron_leggings', 'fishing_rod', 'cooked_mutton', 'spruce_planks', 'gunpowder', 'amethyst_shard', 'chest', 'string', 'cooked_salmon', 'iron_sword', 'raw_copper', 'crafting_table', 'torch', 'lapis_lazuli', 'iron_pickaxe', 'copper_ingot', 'stone_pickaxe', 'wooden_hoe', 'stick', 'amethyst_block', 'salmon', 'calcite', 'gravel', 'white_bed', 'bone', 'dirt', 'mutton', 'white_wool', 'spyglass', 'coal', 'wooden_pickaxe', 'cod', 'iron_boots', 'lily_pad', 'cobbled_deepslate', 'lightning_rod', 'snowball', 'stone_axe', 'smooth_basalt', 'diorite', 'water_bucket', 'furnace', 'andesite', 'bucket', 'granite', 'shield', 'iron_helmet', 'raw_iron', 'cobblestone', 'spruce_log', 'cooked_cod', 'tripwire_hook', 'stone_hoe', 'iron_chestplate', 'stone_sword';

- **Trial 3**: 'spruce_planks', 'dirt', 'shield', 'redstone', 'clock', 'diamond_sword', 'iron_chestplate', 'stone_pickaxe', 'leather', 'string', 'chicken', 'chest', 'diorite', 'iron_leggings', 'black_wool', 'cobblestone_wall', 'cobblestone', 'cooked_chicken', 'feather', 'stone_sword', 'raw_gold', 'gravel', 'birch_planks', 'coal', 'cobbled_deepslate', 'oak_planks', 'iron_pickaxe', 'granite', 'tuff', 'crafting_table', 'iron_helmet', 'stone_hoe', 'iron_ingot', 'stone_axe', 'birch_boat', 'stick', 'sand', 'bone', 'raw_iron', 'beef', 'rail', 'oak_sapling', 'kelp', 'gold_ingot', 'birch_log', 'wheat_seeds', 'cooked_mutton', 'furnace', 'arrow', 'stone_shovel', 'white_wool', 'andesite', 'jungle_slab', 'mutton', 'iron_sword', 'copper_ingot', 'diamond', 'torch', 'oak_log', 'cooked_beef', 'copper_block', 'flint', 'bone_meal', 'raw_copper', 'wooden_pickaxe', 'iron_boots', 'wooden_sword'.

The items collected by ReAct (Yao et al., 2022) in each trial is

- **Trial 1**: 'bamboo', 'dirt', 'sand', 'wheat_seeds';

- **Trial 2**: 'dirt', 'rabbit', 'spruce_log', 'spruce_sapling';

- **Trial 3**: 'dirt', 'pointed_dripstone';

The items collected by Reflexion (Shinn et al., 2023) in each trial is

- **Trial 1**: 'crafting_table', 'orange_tulip', 'oak_planks', 'oak_log', 'dirt';

- **Trial 2**: 'spruce_log', 'dirt', 'clay_ball', 'sand', 'gravel';

- **Trial 3**: 'wheat_seeds', 'oak_log', 'dirt', 'birch_log', 'sand'.

The items collected by AutoGPT (Richards, 2023) in each trial is

- **Trial 1**: 'feather', 'oak_log', 'leather', 'stick', 'porkchop', 'chicken', 'crafting_table', 'wheat_seeds', 'oak_planks', 'dirt', 'mutton';

- **Trial 2**: 'wooden_pickaxe', 'iron_ingot', 'stone', 'coal', 'spruce_planks', 'string', 'raw_copper', 'crafting_table', 'diorite', 'andesite', 'furnace', 'torch', 'spruce_sapling', 'granite', 'iron_pickaxe', 'stone_pickaxe', 'wooden_axe', 'raw_iron', 'stick', 'spruce_log', 'dirt', 'cobblestone';

- **Trial 3**: 'wooden_shovel', 'wooden_pickaxe', 'iron_ingot', 'stone', 'cod', 'coal', 'oak_log', 'flint', 'raw_copper', 'crafting_table', 'diorite', 'furnace', 'andesite', 'torch', 'granite', 'lapis_lazuli', 'iron_pickaxe', 'stone_pickaxe', 'raw_iron', 'stick', 'gravel', 'oak_planks', 'dirt', 'iron_axe', 'cobblestone'.

### B.4.2 Extensive Map Traversal

Agent trajectories for map coverage are displayed in Fig. 7.

The terrains traversed by VOYAGER in each trial is

- **Trial 1**: 'meadow', 'desert', 'river', 'savanna', 'forest', 'plains', 'bamboo_jungle', 'dripstone_caves';

- **Trial 2**: 'snowy_plains', 'frozen_river', 'dripstone_caves', 'snowy_taiga', 'beach';

- **Trial 3**: 'flower_forest', 'meadow', 'old_growth_birch_forest', 'snowy_slopes', 'frozen_peaks', 'forest', 'river', 'beach', 'ocean', 'sunflower_plains', 'plains', 'stony_shore'.

The terrains traversed by ReAct (Yao et al., 2022) in each trial is

- **Trial 1**: 'plains', 'desert', 'jungle';

- **Trial 2**: 'snowy_plains', 'snowy_taiga', 'snowy_slopes';

- **Trial 3**: 'dark_forest', 'dripstone_caves', 'grove', 'jagged_peaks'.

The terrains traversed by Reflexion (Shinn et al., 2023) in each trial is

- **Trial 1**: 'plains', 'flower_forest';

- **Trial 2**: 'snowy_taiga';

- **Trial 3**: 'old_growth_birch_forest', 'river', 'ocean', 'beach', 'plains'.

The terrains traversed by AutoGPT (Richards, 2023) in each trial is

- **Trial 1**: 'plains', 'dripstone_caves', 'savanna', 'meadow';

- **Trial 2**: 'snowy_taiga';

- **Trial 3**: 'plains', 'stony_shore', 'forest', 'ocean'.

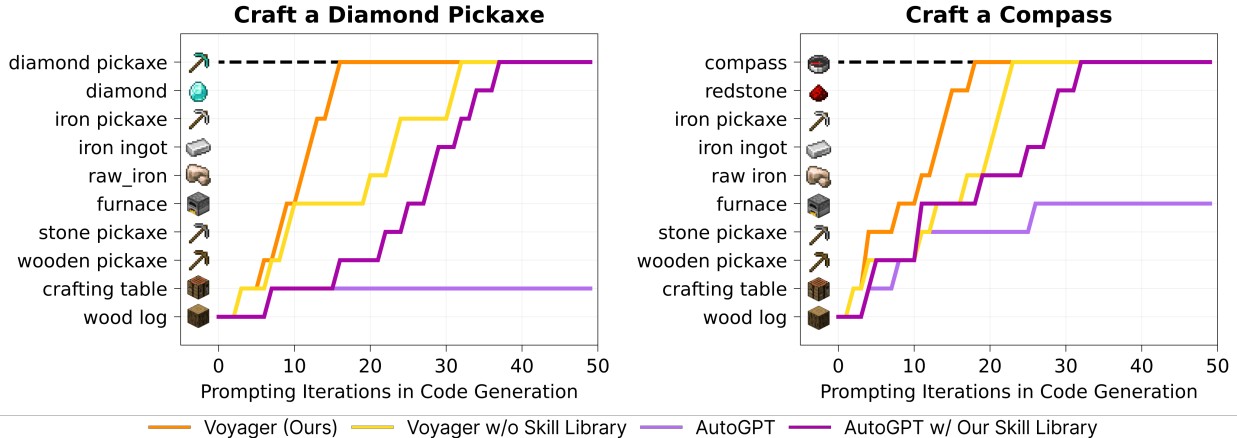

Figure A.2: Zero-shot generalization to unseen tasks. We visualize the intermediate progress of each method on the other two tasks. We do not plot ReAct and Reflexion since they do not make any meaningful progress.

### B.4.3   Efficient Zero-Shot Generalization to Unseen Tasks

The results of zero-shot generalization to unseen tasks for the other two tasks are presented in Fig. A.2. Similar to Fig. 8, VOYAGER consistently solves all tasks, while the baselines are unable to solve any task within 50 prompting iterations. Our skill library, constructed from lifelong learning, not only enhances VOYAGER's performance but also provides a boost to AutoGPT (Richards, 2023).

### B.4.4   Accurate Skill Retrieval

We conduct an evaluation of our skill retrieval (309 samples in total) and the results are in Table. A.4. The top-5 accuracy standing at 96.5% suggests our retrieval process is reliable (note that we include the top-5 relevant skills in the prompt for synthesizing a new skill).

Table A.4: Skill retrieval accuracy.

| Top-1 Acc | Top-2 Acc | Top-3 Acc | Top-4 Acc | Top-5 Acc |
|---|---|---|---|---|
| $80.2 \pm 3.0$ | $89.3 \pm 1.8$ | $93.2 \pm 0.7$ | $95.2 \pm 1.8$ | $96.5 \pm 0.3$ |

### B.4.5   Robust to Model Variations

In the main paper, all of Voyager's experiments are conducted with `gpt-4-0314`. We additionally run new experiments with `gpt-4-0613` and find that the performance is roughly the same (Fig. A.3). It demonstrates that Voyager is robust to model variations.

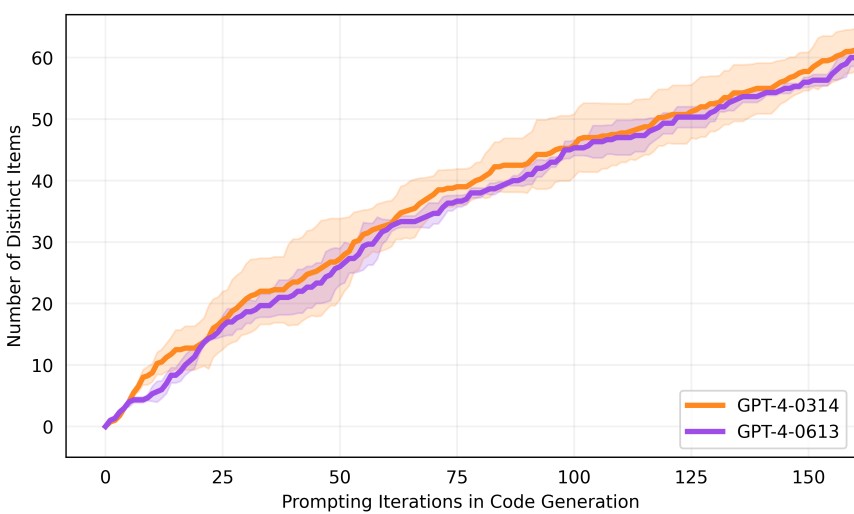

Figure A.3: Voyager's performance with GPT-4-0314 and GPT-4-0613.

