# OpenReview forum: "Voyager: An Open-Ended Embodied Agent with Large Language Models"
_TMLR — Accepted by TMLR_

### Review · Reviewer_eudD · 2024-01-01

**Summary Of Contributions:**

The paper presents Voyager, a novel framework for autonomously training an agent to play Minecraft. Voyager uses a curriculum learning (CL) approach: it keeps exploring the world and builds a library of skills, that can be retrieved to solve new tasks. The major innovation of this framework is the aid of a "prompting mechanism" that eliminates the needs of human intervention.
Voyager shows impressive results and beats all SOTA competitors.

The paper is well written, results are clear, and the proposed framework is interesting.
To the best of my knowledge, while the use of CL and a library of skills is not new to this type of problems, the combination with LLMs makes Voyager novel.

In particular, Voyager agent's actions are code routine: if the action is "craft axe" then Voyager outputs JavaScript code that produces low-level commands that try to craft an axe. If the code fails, Voyager receives some feedback from the game (e.g., "not enough material X to craft an axe") and refines the code. If the task is successful, the agent saves the code as skill for future reference. This way, one task at the time, it builds a library of skills of increasing difficulty.
The way skills are learned, in fact, are constrained to the higher goal of "discover as many diverse things as possible" and that "the task should not be too hard". This is done to reflect the behavior of human players, who start learning basic tasks and then move on to harder tasks.

All this reasoning is done by Chat-GPT. First, information about the current state of the game (e.g., inventory, biome, time, health) are passed to Chat-GPT. In turn, Chat-GPT decides what to do (find food, harvest materials, craft items, ...) and outputs code that performs actions in Minecraft.

**Audience:**

Yes

**Broader Impact Concerns:**

Discussed in "strength and weaknesses".

**Claims And Evidence:**

Yes

**Requested Changes:**

Discussed in "strengths and weaknesses".

**Strengths And Weaknesses:**

Results are convincing and the authors also perform some ablation to investigate components of Voyager (exploration, CL, and self-verification).
However, being not very familiar with this kind of work, it is hard for me to assess its novelty and where it places itself with respect to current literature.

It seems to me that all the reasoning is done by Chat-GPT, and the Voyager framework simply provides Chat-GPT some constrains and guidance ("learn diverse tasks" and "start with simple tasks"). In the end, Voyager does not really learn, Chat-GPT does.
The authors often argue that they want to mimic human learning, so Voyager starts with simple tasks and then builds a library of skills over time. Yet, the learning that happens does not look like human learning to me.
Rather than being like a new human player learning Minecraft autonomously for the first time all by itself, Voyager seems a new player being instructed step-by-step by a pro player (Chat-GPT). Rather than really learning, one could argue that the player just follows instructions blindly.
Overall, this seems more like a new way to use Chat-GPT rather than actually showing a new way of learning.
I'd appreciate if the authors could elaborate on this.

Having more discussion on the baseline would also help understanding the impact of Voyager. I am not familiar with ReAct, Reflection, and AutoGPT, and their very brief description provided by the authors is not sufficient to understand the difference between them, and how Voyager is better.

Finally, how can Voyager be applied to other domains, and impact other research? The authors briefly mention "robotics" at the end of the paper but more discussion on it would help. I find it hard to believe that Chat-GPT would help in real-world taks where very little data is available, and where maybe simulation is impossible or inaccurate.

---

> ### Author Response · Authors · 2024-01-21
>
> Thank you for your thoughtful comments and feedback. We are thrilled that you find our paper well-written, our results clear and convincing, and our proposed framework interesting. We address your additional concerns below:
>
> > Voyager does not really learn, Chat-GPT does … Rather than being like a new human player learning Minecraft autonomously for the first time all by itself, Voyager seems a new player being instructed step-by-step by a pro player (Chat-GPT) … Overall, this seems more like a new way to use Chat-GPT rather than actually showing a new way of learning.
>
> We would like to clarify the role of GPT-4. Instead of being a “pro player”, GPT-4 is an LLM that does not have open-ended exploration and Minecraft-playing skills out of the box. It only serves as a base reasoning engine upon which AI agent algorithms can be built. Voyager is our proposal of an advanced LLM-based agent algorithm with only blackbox access to GPT-4. Previously, ReAct [1], Reflexion [2], and AutoGPT [3] are all in the same category, but they significantly underperform Voyager as our experiments demonstrate.
>
> Voyager also introduces a novel kind of learning. It iteratively constructs a codebase of skills as it keeps exploring in the open-ended world. We can think of the codebase as an explicit form of “learned parameters”, acquired by coding instead of gradient descent. Even though Voyager does not modify the underlying GPT-4, it is still able to accumulate new skills and improve itself continuously, hence the term “in-context lifelong learning capability” as stated in our abstract.
>
> > Having more discussion on the baseline would also help understanding the impact of Voyager. I am not familiar with ReAct, Reflection, and AutoGPT, and their very brief description provided by the authors is not sufficient to understand the difference between them, and how Voyager is better.
>
> We provide additional information on these baselines in Appendix Sec C.2 and Table A.3. We present the content of Table A.3 below for better visibility.
>
> |                                      | ReAct [1] | Reflexion [2] | AutoGPT [3] | Voyager |
> |------------------------------|:---------:|:-------------:|:-----------:|:-------:|
> | Chain-of-Thought [4]     |       ✓       |          ✓         |          ✓         |      ✓      |
> | Self Verification             |                 |          ✓         |                      |      ✓      |
> | Environment Feedback |       ✓       |          ✓         |          ✓          |      ✓      |
> | Execution Errors            |                |          ✓         |           ✓         |      ✓      |
> | Agent State                    |       ✓      |          ✓         |           ✓         |      ✓      |
> | Skill Library                    |                |                      |                       |      ✓      |
> | Automatic Curriculum    |                |                      |                       |      ✓      |
>
> It is clear to see Voyager is the only method that incorporates all the components that work in tandem to deliver the best performance, which we believe are valuable insights that contribute to the community.

---

> > ### Author Response · Authors · 2024-01-21
> >
> > > Finally, how can Voyager be applied to other domains, and impact other research? The authors briefly mention "robotics" at the end of the paper but more discussion on it would help. I find it hard to believe that Chat-GPT would help in real-world tasks where very little data is available, and where maybe simulation is impossible or inaccurate.
> >
> > Good point! Voyager's method is general-purpose and does not make strong assumptions about the environment. In principle, it can be adapted to other domains where code serves as the action space and comes with a well-explained API, such as Webshop [5] and WebArena [6].
> >
> > In addition, prior works on robotic manipulation such as Code as Policies [7] and ProgPrompt [8] also assume there exists a skill library or control primitives (pick and place an object, open a container, switch on/off an object, etc.) and use LLMs to generate code for policy execution. Since LLMs have a lot of common sense about real-world knowledge, it would be interesting to see how LLMs can expand the skill library based on the given control primitives and previously learned skills to solve robotic manipulation tasks in the real world. Let’s say we tell LLMs: “You are a cooking expert. Here are the skills that I already acquired: find a microwave, put an object into a microwave, open and close a microwave. Please propose a novel task for me to pursue.” It would make sense for LLMs to output something like “microwave some food” as the next task and write a program to compose the previously learned skills to synthesize a new one through iterative prompting.
> >
> > For domains less familar to LLMs, we can use LLMs to reference knowledge from wiki pages or handbooks (e.g., MineDojo [9] compiles a wiki dataset for Minecraft). This summarized information can provide valuable insights into how the domain works and therefore guide LLMs in making decisions and identifying the next steps in uncharted environments.
> >
> >
> > [1] Yao et al., ReAct: Synergizing Reasoning and Acting in Language Models, ICLR 2023.
> >
> > [2] Shinn et al., Reflexion: Language Agents with Verbal Reinforcement Learning, NeurIPS 2023.
> >
> > [3] Toran Bruce Richards. Significant-gravitas/auto-gpt: An experimental open-source attempt to make gpt-4 fully autonomous., 2023. URL https://github.com/Significant-Gravitas/Auto-GPT/tree/master.
> >
> > [4] Wei et al., Chain-of-Thought Prompting Elicits Reasoning in Large Language Models, NeurIPS 2022.
> >
> > [5] Yao et al., WebShop: Towards Scalable Real-World Web Interaction with Grounded Language Agents, NeurIPS 2022.
> >
> > [6] Zhou et al., WebArena: A Realistic Web Environment for Building Autonomous Agents, 2023.
> >
> > [7] Liang et al., Code as Policies: Language Model Programs for Embodied Control, ICRA 2023.
> >
> > [8] Singh et al., ProgPrompt: Generating Situated Robot Task Plans using Large Language Models, ICRA 2023.
> >
> > [9] Fan et al., MineDojo: Building Open-Ended Embodied Agents with Internet-Scale Knowledge, NeurIPS 2022.

---

### Review · Reviewer_HygD · 2024-01-03

**Summary Of Contributions:**

This paper proposes a system that acts as an agent in Minecraft, which progressively acquires and represents the complex action and state space of the environment through exploration and abstraction. The main contributions of the approach are: (a) a curriculum generated from an LLM to suggest new subgoals to pursue at a particular environment state, (b) an iterative prompting mechanism where the goal is to write a piece of executable code representing this subgoal, and a (c) skill library that stores previously-generated pieces of code for later use. Experiments are performed on the Minecraft environment, and comparisons are made with existing text-based approaches that focus on self-reflection / inference-time refinement for (a) iterative refinement / critique at inference time and (b) inference-time policies that decompose the current goal into subgoals.

**Audience:**

Yes

**Claims And Evidence:**

Yes

**Requested Changes:**

- Discussion on how priors encoded in pretrained LMs may not be available for new games, thus the exploration phase / automatic curriculum may not transfer as well
- Experiments using the full tech tree to create a curriculum instead of a small number of hand-picked subgoals as the manual curriculum

**Strengths And Weaknesses:**

Strengths:
- The approach is an elegant composition of different ways of using LLMs that form a clearly strong game agent.
- The experimental setup is really thorough, and the adaptation of existing approaches (ReAct, Reflexion, AutoGPT) providing access to the skill library etc. are well-tested.
- Overall, the experiments are sufficient to show that this approach is very strong!

Weaknesses:
- My main concern is that one critical component of this approach, the automatic curriculum, is useful primarily because it's taking advantage of existing knowledge of the Minecraft tech tree and game strategies that are encoded in GPT-4. This is not a fundamental flaw, but it would be interesting to discuss more how the proposed approach would generalize to completely new games without mentions on the web that have equally complex tech trees.
- The manual curriculum seems quite small and focused on a single goal in the tech tree (mining a diamond). Can't a larger manual curriculum be constructed automatically from the tech tree itself, e.g. by flattening it into some sequence?

Questions:
- In Figure 5 left, it seems like the lines describing how to create a stick could be factored out into their own function, right? Since 2 sticks are being created (also, it seems like this code is only creating 1 stick even though 2 more are needed?). Where, if anywhere, in this pipeline is this kind of refactoring taking place? Or would it only be possible if there was already a function called craftStick available in the skill library?
- In the zero-shot generalization experiment, does the evaluation of Voyager also allow for exploration and adding to the skill library, or are these two aspects fixed?
- I'd like to see more details on what "self-ask questions" look like in Section 2.1.
- The human feedback in Fig 10 is type (1) right, using humans as a critic?

---

> ### Author Response · Authors · 2024-01-21
>
> Thank you so much for your thoughtful feedback and positive review. We are thrilled to hear your positive remarks on Voyager, recognizing its elegant composition of different components, thorough experiments, and strong performance. We have also run additional experiments to address your concerns. Below are our responses to your points:
>
> > It would be interesting to discuss more how the proposed approach would generalize to completely new games without mentions on the web that have equally complex tech trees.
>
> Great point. We can use GPT-4 to extract game mechanics from wiki pages or handbooks. This summarized information can provide valuable insights into how the domain works, which is especially helpful in unfamiliar environments. In addition, Voyager requires environment observation APIs and low-level control primitives to be explained/demonstrated within the LLM’s prompt, like Inner Monologue [1]  and Code as Policies [2].
>
> > Can't a larger manual curriculum be constructed automatically from the tech tree itself, e.g. by flattening it into some sequence?
>
> Thanks for your suggestion. We want to highlight that our automatic curriculum is designed to propose suitable and useful tasks based on its current skill level and world state, while a manual curriculum requires access to the full tech tree and is not scalable for open-ended exploration. We have expanded the manual curriculum by incorporating additional tasks derived from the tech tree, and the results are shown in Fig. 9 Left. It is important to note that the manual curriculum is not aware of the current skill level and world state, limiting its ability to consistently recommend the most pertinent tasks at any given moment.
>
> > In Fig. 5 Left, it seems like the lines describing how to create a stick could be factored out into their own function … It seems like this code is only creating 1 stick even though 2 more are needed? … Where, if anywhere, in this pipeline is this kind of refactoring taking place?
>
> To clarify, the third parameter in the `craftItem` function specifies the number of iterations for the recipe execution. In Minecraft, executing the stick recipe once consumes 2 logs and yields 4 sticks, which exceeds the required amount of 2.
>
> Voyager does not engage in explicit code refactoring. However, it is conceivable that a repetitive task (but in a varied world state) might be suggested by the automated curriculum. In such scenarios, the old skill could be overwritten by a new one.
>
>
> > In the zero-shot generalization experiment, does the evaluation of Voyager also allow for exploration and adding to the skill library, or are these two aspects fixed?
>
> The skill library is fixed as we aim to conduct a controlled study to demonstrate the effectiveness of our "pre-trained" skill library for unseen tasks. Also, we utilize GPT-4 to break down the unseen task into a series of subgoals, so we are not doing open-ended exploration in this particular study.
>
>
> > I'd like to see more details on what "self-ask questions" look like in Section 2.1.
>
> Sure. We inform GPT-3.5 about the agent's current state, along with a history of tasks completed and those that are unsuccessful. Then, we instruct GPT-3.5 to formulate a set of questions. Each question must be directly related to a specific concept in Minecraft. For instance, "How do you make a wooden pickaxe?" Additionally, the questions should be self-explanatory. An example of such a question is, "What are the blocks that can be discovered in a sparse jungle?" The responses to these questions, provided by GPT-3.5, then serve as additional context for brainstorming new tasks.
>
>
> > The human feedback in Fig 10 is type (1) right, using humans as a critic?
>
> The Nether Portal example uses a human as a critic: “`Your portal does not have the correct size. For a 10-obsidian portal, the top and bottom edges should only have 2 blocks, and the two-side edge should have 3 blocks. The 4 corners of the portal should use another material such as cobblestone.`”
>
> The house example uses a human as a curriculum. We guide Voyager step-by-step, from laying the foundation to adding windows and a door:
>
>     1. “First, don't think of any other things, you only need to build the main body using oak_log with size 5x5 and 3-block height. Start from offset Y = 0 level.”
>
>     2. “You just built a 5*5 shelter starting at (615, 79, 1206). Now you need to add some wood stairs above the shelter as decoration. Place 9 stairs on each corner, the center of each edge, and the center of the building.”
>
>     3. “You just built a 5*5 shelter starting at (615, 79, 1206). You need to add 8 windows and a door. The door should be in the middle of one of the walls. The windows should be on each wall, with 1 block above the ground and 1 block between them.”
>
>
> [1] Huang et al., Inner Monologue: Embodied Reasoning through Planning with Language Models, CoRL 2022.
>
> [2] Liang et al., Code as Policies: Language Model Programs for Embodied Control, ICRA 2023.

---

### Review · Reviewer_ogTF · 2024-01-15

**Summary Of Contributions:**

This paper presents an embodied lifelong learning agent in Minecraft and explores the world, acquires and reuses skills. It consists of three components:
* an automatic curriculum that proposes novel tasks
* a skill library for storing and retrieving new skills
* an iterative prompting mechanism

Voyager interacts with Minecraft via code and the high-level Mineflayer API. All the different components are implemented with black box queries to GPT-4. The methods leads to quite advance and complex minecraft behaviour. It performs better than reasonable LLM-based baselines in a number of metrics. Futhermore the paper presents ablation studies in which parts of the overall system are removed. The automatic curriculum is found to be crucial and the skill library appears to help avoid plateauing. Finally GPT-4 is found to outperform GPT-3.5 in code generation.

**Audience:**

Yes

**Broader Impact Concerns:**

None.

**Claims And Evidence:**

Yes

**Requested Changes:**

My main concern is that the main text lacks some details that would make it easier to read as a self-contained document.
* Consider moving the algorithm in pseudo code to the main text.
* Replace figure 7 with the similar figure A.2 from the appendix that shows paths instead of circles. As it stands there is not enough information to interpret fig 7. Please highlight the starting position in some way.
* please give further details on figure 1: What do the shaded regions represent? I am assuming there are several underlying runs shown. How are the shown items related to only one of the runs? In particular this presentation may lead a reader to conclude that the diamond tool is always found. If I understand table 1 correctly, this is not the case?

**Strengths And Weaknesses:**

Strengths:
* Very impressive overall performance. It is surprisingly and instructive to see how much can be achieved purely with black box queries to a SOTA LLM.
* Careful and insightful ablation experiments.

Weaknesses:
* The high level action API abstracts away a large part minecraft. It remains to be seen to what extent the same performance can be obtained in other domains.
* The paper is relatively clear but I think the overall presentation could be improved. See suggestions below.

---

> ### Author Response · Authors · 2024-01-21
>
> Thank you so much for your constructive feedback and positive review. We are glad that you find our performance impressive, our method instructive, and our ablation experiments careful and insightful. We answer your questions below and have incorporated your feedback into our revised version.
>
> > The high level action API abstracts away a large part minecraft.
>
> We follow prior works like Code as Policies [1] and Inner Monologue [2] to equip LLMs with control primitives. Our work’s focus is on pushing the limits of LLMs for lifelong embodied agent learning, rather than solving the 3D perception or sensorimotor control problems. Voyager is orthogonal and can be combined with gradient-based approaches like VPT [3] as long as the controller provides a code API.
>
>
> > It remains to be seen to what extent the same performance can be obtained in other domains.
>
> We appreciate the reviewer's suggestion to investigate the performance of our method in other domains.
>
> Voyager's method is general-purpose enough and does not make strong assumptions about the environment. In principle, it can be adapted to other domains where code serves as the action space and comes with a well-explained API, such as Webshop [4] and WebArena [5]. We plan to extend it to other environments in future work.
>
> We choose to focus on Minecraft because it is a highly representative environment that requires long-horizon planning, compositional reasoning, and extensive exploration. Prior works like VPT [3], MineDojo [6], and DECKARD [7] only study Minecraft as well, and AlphaGo [8], OpenAI Five [9], and AlphaStar [10] only study one game each. All of these works contribute valuable insights to the larger policy learning community, which we believe Voyager does too.
>
> > Consider moving the algorithm in pseudo code to the main text.
>
> Good point! We have moved it to the main text.
>
> > Replace Figure 7 with the similar Figure A.2 … Please highlight the starting position in some way.
>
> Thanks for the comment! We have revised Figure 7 according to your suggestion.
>
> > Please give further details on Figure 1: What do the shaded regions represent? … How are the shown items related to only one of the runs? In particular this presentation may lead a reader to conclude that the diamond tool is always found.
>
> The shaded region represents the standard deviation of the items acquired by each agent over three experimental runs. We focus on highlighting the significant and noteworthy items obtained by each agent. Notably, our Voyager is the only method that can obtain a diamond tool in one of the 3 runs. We have clarified this in the caption of Figure 1.
>
> [1] Liang et al., Code as Policies: Language Model Programs for Embodied Control, ICRA 2023.
>
> [2] Huang et al., Inner Monologue: Embodied Reasoning through Planning with Language Models, CoRL 2022.
>
> [3] Baker et al., Video PreTraining (VPT): Learning to Act by Watching Unlabeled Online Videos, NeurIPS 2022.
>
> [4] Yao et al., WebShop: Towards Scalable Real-World Web Interaction with Grounded Language Agents, NeurIPS 2022.
>
> [5] Zhou et al., WebArena: A Realistic Web Environment for Building Autonomous Agents, 2023.
>
> [6] Fan et al., MineDojo: Building Open-Ended Embodied Agents with Internet-Scale Knowledge, NeurIPS 2022.
>
> [7] Nottingham et al., Do Embodied Agents Dream of Pixelated Sheep: Embodied Decision Making using Language Guided World Modelling, ICML 2023.
>
> [8] Silver et al., Mastering Chess and Shogi by Self-Play with a General Reinforcement Learning Algorithm, 2017.
>
> [9] OpenAI et al., Dota 2 with Large Scale Deep Reinforcement Learning, 2019.
>
> [10] Vinyals et al., AlphaStar: Mastering the real-time strategy game StarCraft II, 2019.

---

### Author Response · Authors · 2024-01-21
**Updated Manuscript and Response to All Reviewers**

We sincerely thank all reviewers for their insightful feedback and thoughtful suggestions. We are particularly grateful that many of you acknowledge the novelty of Voyager’s three pillars (an automatic curriculum, an iterative prompting mechanism, and an ever-growing skill library), and the comprehensiveness of our experiments and ablations.

Below, we address the feedback from each reviewer. The paper PDF has been updated with suggested revisions, highlighted in red. We welcome any further discussions!

---

### Decision · Action_Editor_wj2T · 2024-02-18

**Recommendation:** Accept as is

**Comment:**

All the reviewers are in the agreement that the method is sound and impressive. And while it might to be clear how well it might generalize beyond MineCraft, there is value in publishing the paper.

eudD: "The paper combines known and new techniques into a novel framework that trains agents to play Minecraft better than any other existing framework. The paper is well written, resuls are convincing and well discussed, and the authors have addressed all my concerns."
ogTF: "I think this paper shows very impressive performance in Minecraft."
HygD: "While I am still unsure about whether this approach will scale with completely new games that have different dynamics than ones whose relevant documentation / discussion has been heavily included in training data for existing LLMs, I think the approach itself is noteworthy and worthy of acceptance."

**Audience:**

This paper would be of interest to the autonomous agents, and openended community.

**Claims And Evidence:**

A lifelong learning agent consisting of a curriculum, expandable skill library, and environment / feedback-conditioned prompting mechanism to solve open-ended problems. The agent is evaluated on the MineCraft.

All the reviewers agree, and I concur that the results are solid and the claims in the paper are sound.